# Universal Neurons in GPT2 Language Models

**Wes Gurnee**[*]                                                                                                *wesg@mit.edu*
*Massachusetts Institute of Technology*

**Theo Horsley**                                                                                            *tjh203@cam.ac.uk*
*University of Cambridge*

**Zifan Carl Guo**                                                                                          *carlguo@mit.edu*
*Massachusetts Institute of Technology*

**Tara Rezaei Kheirkhah**                                                                                    *tarark@mit.edu*
*Massachusetts Institute of Technology*

**Qinyi Sun**                                                                                            *wendysun@mit.edu*
*Massachusetts Institute of Technology*

**Will Hathaway**                                                                                          *willhath@mit.edu*
*Massachusetts Institute of Technology*

**Neel Nanda**[†]                                                                                    *neelnanda27@gmail.com*

**Dimitris Bertsimas**[†]                                                                                  *dbertsim@mit.edu*
*Massachusetts Institute of Technology*

**Reviewed on OpenReview:** *https://openreview.net/forum?id=ZeI104QZ8I&noteId=k3hwQgKrsg*

## Abstract

A basic question within the emerging field of mechanistic interpretability is the degree to which neural networks learn the same underlying mechanisms. In other words, are neural mechanisms universal across different models? In this work, we study the universality of individual neurons across GPT2 models trained from different initial random seeds, motivated by the hypothesis that universal neurons are likely to be interpretable. In particular, we compute pairwise correlations of neuron activations over 100 million tokens for every neuron pair across five different seeds and find that 1-5% of neurons are universal, that is, pairs of neurons which consistently activate on the same inputs. We then study these universal neurons in detail, finding that they usually have clear interpretations and taxonomize them into a small number of neuron families. We conclude by studying patterns in neuron weights to establish several universal functional roles of neurons in simple circuits: deactivating attention heads, changing the entropy of the next token distribution, and predicting the next token to (not) be within a particular set.

## 1 Introduction

As large language models (LLMs) become more widely deployed in high-stakes settings, our lack of understanding of why or how models make decisions creates many potential vulnerabilities and risks (Bommasani et al., 2021; Hendrycks et al., 2023; Bengio et al., 2023). While some claim deep learning based systems are fundamentally inscrutable, artificial neural networks seem unusually amenable to empirical science compared to other complex systems: they are fully observable, (mostly) deterministic, created by processes we control,

---

[*]Corresponding Author; [†] Senior Author

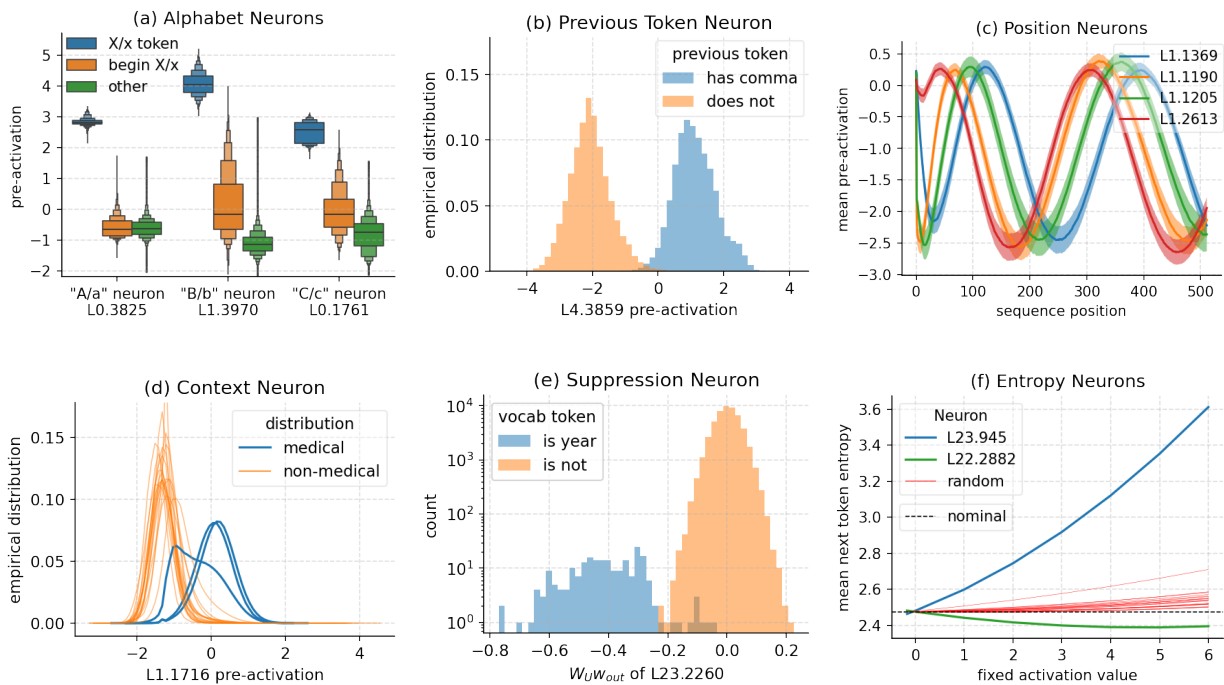

Figure 1: Universal neurons in GPT2 models, interpreted via their activations (a-d), weights (e), and causal interventions (f). (a) Neurons which activate primarily on a specific individual letter and secondarily on tokens which begin with the letter; (b) Neuron which activates approximately if and only if the previous token contains a comma; (c) Neurons which activate as a function of absolute token position in the context (shaded area denotes standard deviation around the mean); (d) A neuron which activates in medical contexts (e.g. pubmed abstracts) but not in non-medical distributions; (e) a neuron which decreases the probability of predicting any integer tokens between 1700 and 2050 (i.e., years); (f) Neurons which change the entropy of the next token distribution when causally intervened.

admit complete mathematical descriptions of their form and function, can be run on any input with arbitrary modifications made to their internals, all at low cost and on computational timescales (Olah, 2021). An advanced science of interpretability enables a more informed discussion of the risks posed by advanced AI systems and lays firmer ground to engineer systems less likely to cause harm (Doshi-Velez and Kim, 2017; Bender et al., 2021; Weidinger et al., 2022; Ngo et al., 2023; Carlsmith, 2023).

Olah et al. (2020b) propose three speculative claims regarding the interpretation of artificial neural networks: that features—directions in activation space representing properites of the input—are the fundamental unit of analysis, that features are connected into circuits via network weights, and that features and circuits are universal across models. That is, analogous features and circuits form in a diverse array of models and that different training trajectories converge on similar solutions (Li et al., 2015). Taken seriously, these hypotheses suggest a strategy for discovering important features and circuits: look for that which is universal. This line of reasoning motivates our work, where we leverage different notions of universality to identify and study individual neurons that represent features or underlie circuits.

Beyond discovery, the degree to which neural mechanisms are universal is a basic open question that informs what kinds of interpretability research are most likely to be tractable and important. If the universality hypothesis is largely true in practice, we would expect detailed mechanistic analyses (Cammarata et al., 2021; Wang et al., 2022a; Olsson et al., 2022; Nanda et al., 2023; McDougall et al., 2023) to generalize across models such that it might be possible to develop a periodic table of neural circuits which can be automatically referenced when interpreting new models (Olah et al., 2020b). Conversely, it becomes less sensible to dedicate substantial manual labor to understand low-level details of circuits if they are completely different in every

model, and instead more efficient to allocate effort to engineering scalable and automated methods that can aid in understanding and monitoring higher-level representations of particular interest (Burns et al., 2022; Conmy et al., 2023; Bills et al., 2023; Zou et al., 2023; Bricken et al., 2023). However, even in the case that not all features or circuits are universal, those which are common across models are likely to be more fundamental (Bau et al., 2018; Olsson et al., 2022), and studying them should be prioritized accordingly.

In this work, we study the universality of individual neurons across GPT2 language models (Radford et al., 2019) trained from five different random initializations (Karamcheti et al., 2021). While it is well known that individual neurons are often polysemantic (Nguyen et al., 2016; Olah et al., 2020b; Elhage et al., 2022b; Gurnee et al., 2023) i.e., represent multiple unrelated concepts, we hypothesized that universal neurons were more likely to be monosemantic (see §A.1), potentially giving an approximation on the number of independently meaningful neurons. We choose to study models of the same architecture trained on the same data to have the most favorable experimental conditions for measuring universality to establish a rough bound for the universality over larger changes. We begin by operationalizing neuron universality in terms of activation correlations, that is, whether there exist pairs of neurons across different models which consistently activate on the same inputs. We compute pairwise correlations of neuron activations over 100 million tokens for every neuron pair across the different seeds and find that only 1-5% of neurons pass a target threshold of universality compared to random baselines (§ 4.1). We then study these universal neurons in detail, analyzing various statistical properties of both weights and activations (§ 4.2), and find that they usually have clear interpretations and taxonomize them into a small number of neuron families (§ 4.3).

In Section 5 we study a more abstract form of universality in terms of neuron weights rather than activations. That is, rather than understand a neuron in terms of the inputs which cause it to activate, understand a neuron in terms of the effects the neuron has on later model components or directly on the final prediction. Specifically, we analyze patterns in the compositional structure of the weights and find consistent outliers in how neurons affect other network components, constituting very simple circuits. In Section 5.1, we show there exists a large family of late layer neurons which have clear roles in predicting or suppressing a coherent set of tokens (e.g., second-person pronouns or single digit numbers), where the suppression neurons typically come in later layers than the prediction neurons. We then investigate a small set of neurons that leverage the final layer-norm operation to modulate the overall entropy of the next token prediction distribution (§ 5.2). We conclude with an analysis of neurons which control the extent to which an attention head attends to the first token, which empirically controls the output norm of the head, effectively turning a head on or off (§ 5.3).

## 2 Related Work

**Universal Neural Mechanisms** Features and circuits like high-low frequency detectors (Schubert et al., 2021a) and curve circuits (Cammarata et al., 2021) have been found to reoccur in vision models, with some features even reappearing in biological neural networks (Goh et al., 2021). In language models, recent research has found similarly universal circuits and components like induction heads (Olsson et al., 2022) and successor heads (Gould et al., 2023) and that models reuse certain circuit components to implement different tasks (Merullo et al., 2023). There has also been a flurry of recent work on studying more abstract universal mechanisms in language models like function vectors (Todd et al., 2023; Hendel et al., 2023), variable binding mechanisms (Feng and Steinhardt, 2023), and long context retrieval (Variengien and Winsor, 2023). Studying universality in toy models has provided "mixed evidence" on the universality hypothesis (Chughtai et al., 2023) and shown that multiple algorithms exist to implement the same tasks (Zhong et al., 2023; Liao et al., 2023).

**Representational Similarity** Preceding the statement of the universality hypothesis in mechanistic interpretability, there has been substantial work measuring representational similarity (Klabunde et al., 2023). Common methods include canonical correlation analysis-based measures (Raghu et al., 2017; Morcos et al., 2018), alignment-based measures (Hamilton et al., 2018; Ding et al., 2021; Williams et al., 2022; Duong et al., 2023), matrix-based measures (Kornblith et al., 2019; Tang et al., 2020; Shahbazi et al., 2021; Lin, 2022; Boix-Adsera et al., 2022; Godfrey et al., 2023), neighborhood-based measures (Hryniowski and Wong, 2020; Gwilliam and Shrivastava, 2022), topology-based measures (Khrulkov and Oseledets, 2018; Barannikov et al., 2022), and descriptive statistics (Wang and Isola, 2022; Lu et al., 2022; Lange et al., 2022). Previous work,

mostly in vision models, has yielded mixed conclusions on whether networks with the same architecture learn similar representations. Some studies have found that networks with different initializations "exhibit very low similarity" (Wang et al., 2018) and "do not converge to a unique basis" (Brown et al., 2023), while others have shown that networks learn the same low-dimensional subspaces but not identical basis vectors (Li et al., 2016) and that different models can be linearly stitched together with minimal loss suggesting they learn similar representations (Bansal et al., 2021).

**Analyzing Individual Neurons**   Many prior interpretability studies have analyzed individual neurons. In vision models, researchers have found neurons which activate for specific objects (Bau et al., 2020), curves at specific orientations (Cammarata et al., 2021), high frequency boundaries (Schubert et al., 2021b), multimodal concepts (Goh et al., 2021), as well as for facets (Nguyen et al., 2016) and compositions (Mu and Andreas, 2020) thereof. Moreover, many of these neurons seem universal across models Dravid et al. (2023). In language models, neurons have been found to correspond to sentiment (Radford et al., 2017; Donnelly and Roegiest, 2019), knowledge (Dai et al., 2021), skills (Wang et al., 2022b), de-/re-tokenization (Elhage et al., 2022a), contexts (Gurnee et al., 2023; Bills et al., 2023), position (Voita et al., 2023), space and time (Gurnee and Tegmark, 2023), and many other linguistic and grammatical features (Bau et al., 2018; Xin et al., 2019; Dalvi et al., 2019; 2020; Durrani et al., 2022; Sajjad et al., 2022). More generally, it is hypothesized that neurons in language models form key-value stores (Geva et al., 2020) that facilitate next token prediction by promoting concepts in the vocabulary space (Geva et al., 2022). However, many challenges exist in studying individual neurons, especially in drawing causal conclusions (Antverg and Belinkov, 2021; Huang et al., 2023).

## 3   Conceptual and Empirical Preliminaries

### 3.1   Universality

**Notions of Universality**   Universality can refer to many different notions of similarity, each at a different level of abstraction and with differing measures and methodologies. Similar to Marr's levels of analysis in neuroscience (Hamrick and Mohamed, 2020; Marr, 2010), relevant notions of universality are: *computational* or *functional* universality regarding whether a (sub)network implements a particular input-output-behavior (e.g., whether the next token predictions for two different networks are the same); *algorithmic* universality regarding whether or not a particular function is implemented using the same computational steps (e.g., whether a transformer trained to sort strings always learns the same sorting algorithm); *representational* universality, or the degree of similarity of the information contained within different representations (Kornblith et al., 2019) (e.g., whether every network represents absolute position in the context); and finally *implementation* universality, i.e., whether individual model components learned by different models implement the same specialized computations (e.g., induction heads (Olsson et al., 2022), successor heads (Gould et al., 2023), French neurons (Gurnee et al., 2023), *inter alia*). None of these notions of universality are usually binary, and the universality between components or computations can range from being formally isomorphic to simply sharing a common high-level conceptual or statistical motif.

In this work, we are primarily concerned with implementation universality in the form of whether individual neurons learn to specialize and activate for the same inputs across models. If such universal neurons do exist, then this is also a simple form of functional universality, as the distinct neurons constitute the final node of distinct subnetworks which compute the same output.

**Dimensions of Variations**   Universality must be measured over some independent dimension of variation, i.e., some change in the model, data or, training. For example, model variables include random seed, model size, hyperparameters, and architectural changes; data variables include the data size, ordering, and distribution contents; training variables include loss function, optimizer, regularization, finetuning, and hyperparameters thereof. Assuming that changing random seed is the smallest change, this work primarily focuses on initialization universality in an attempt to bound the expected similarity of larger changes.

### 3.2 Models

We restrict our scope to transformer-based auto-regressive language models (Radford et al., 2018) that currently power the most capable AI systems (Bubeck et al., 2023). Given an input sequence of tokens $x = [x_1, \ldots, x_t] \in \mathcal{X} \subseteq \mathcal{V}^t$ from the vocabulary $\mathcal{V}$, a language model $\mathcal{M} : \mathcal{X} \rightarrow \mathcal{Y}$ outputs a probability distribution over the vocabulary to predict the next token in the sequence.

We focus on a replication of the GPT2 series of models (Radford et al., 2019) with some supporting experiments on the Pythia family (Biderman et al., 2023). For a GPT2-small and GPT2-medium architecture (see § B.3 for hyperparameters) we study five models trained from different random seeds, referred to as GPT2-{small, medium}-[a-e] (Karamcheti et al., 2021).

**Anatomy of a Neuron**  Of particular importance to this investigation is the functional form of the neurons in the feed forward (also known as multi-layer perceptron (MLP)) layers in the transformer. The output of an MLP layer given a normalized hidden state $\mathbf{x} \in \mathbb{R}^{d_{\text{model}}}$ is

$$\text{MLP}(\mathbf{x}) = \mathbf{W}_{\text{out}}\sigma(\mathbf{W}_{\text{in}}\mathbf{x} + \mathbf{b}_{\text{in}}) + \mathbf{b}_{\text{out}} \tag{1}$$

where $\mathbf{W}_{\text{out}}^T, \mathbf{W}_{\text{in}} \in \mathbb{R}^{d_{\text{mlp}} \times d_{\text{model}}}$ are learned weight matrices, $\mathbf{b}_{\text{in}}$ and $\mathbf{b}_{\text{out}}$ are learned biases, and $\sigma$ is an elementwise nonlinear activation function. For all models we study, $\sigma$ is the GeLU activation function $\sigma(\mathbf{x}) = \mathbf{x}\Phi(\mathbf{x})$ (Hendrycks and Gimpel, 2016). One can analyze an individual neuron $j$ in terms of its activation $\sigma(\mathbf{w}_{\text{in}}^j\mathbf{x} + b_{\text{in}}^j)$ for different inputs $\mathbf{x}$, or its weights—row $j$ of $\mathbf{W}_{\text{in}}$ or $\mathbf{W}_{\text{out}}^T$ which respectively dictate for what inputs a neuron activates and what effects it has downstream.

We refer the reader to (Elhage et al., 2021) for a full description of the transformer architecture. We employ standard weight preprocessing techniques described further in B.1.

## 4 The Search for Universal Neurons

### 4.1 How Universal are Individual Neurons?

**Experiment**  Inspired by prior work studying common neurons in neural networks (Li et al., 2015; Bau et al., 2018; Dravid et al., 2023), we compute maximum pairwise correlations of neuron activations across five different models GPT2-{a, b, c, d, e} to find pairs of neurons across models which activate on the same inputs. Let $N(a)$ be the set of neurons in model $a$. For each neuron $i \in N(a)$, we compute the Pearson correlation

$$\rho_{i,j}^{a,m} = \frac{\mathbb{E}\left[(\mathbf{v}^i - \mu_i)(\mathbf{v}^j - \mu_j)\right]}{\sigma_i \sigma_j} \tag{2}$$

with all neurons $j \in N(m)$ in every model $m \in$ {b, c, d, e}, where $\mu_i$ and $\sigma_i$ are the mean and standard deviation of a vector of neuron activations $\mathbf{v}^i$ computed across a dataset of 100 million tokens from the Pile test set (Gao et al., 2020). For a baseline, we also compute $\bar{\rho}_{i,j}^{a,m}$, where instead of taking the correlation of $\rho(\mathbf{v}^i, \mathbf{v}^j)$, we compute $\rho(\mathbf{v}^i, (\mathbf{RV})_j)$ for a random $d_{\text{mlp}} \times d_{\text{mlp}}$ Gaussian matrix $\mathbf{R}$ and the matrix of activations $\mathbf{V}$ for all neurons in a particular layer $N_\ell(m)$. In other words, we compute the correlation between neurons and elements within a random (approximate) rotation of a layer of neurons to establish a baseline correlation for the case where there does not exist a privileged basis (Elhage et al., 2021; Brown et al., 2023) to verify the importance of the neuron basis.

For a set of models $M$ we define the *excess correlation* of neuron $i$ as the difference between the mean maximum correlation across models and the mean maximum baseline correlation in the rotated basis:

$$\varrho_i = \frac{1}{|M|} \sum_{m \in M} \left( \max_{j \in N(m)} \rho_{i,j}^{a,m} - \max_{j \in N_R(m)} \bar{\rho}_{i,j}^{a,m} \right) \tag{3}$$

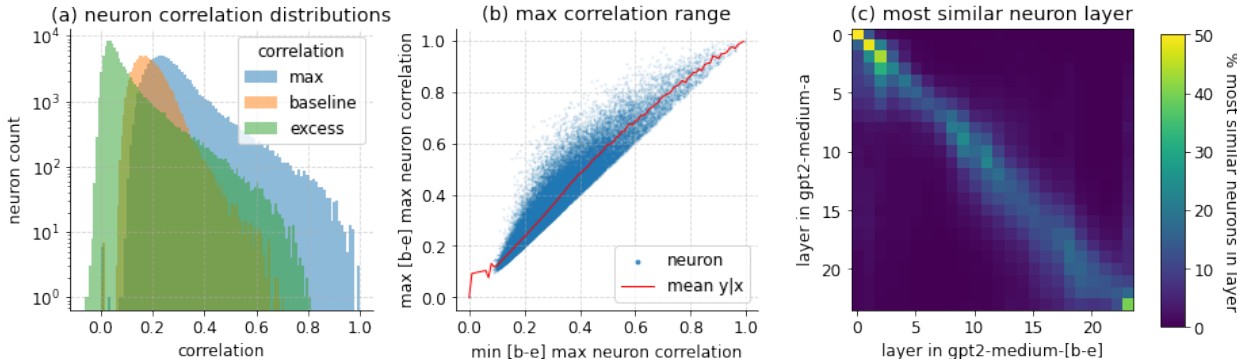

Figure 2: Summary of neuron correlation experiments in GPT2-medium-a. (a) Distribution of the mean (over models b-e) max (over neurons) correlation, the mean baseline correlation, and the difference (excess). (b) The max (over models) max (over neurons) correlation compared to the min (over models) max (over neuron) correlation for each neuron. (c) Percentage of layer pairs with most similar neuron pairs.

**Results** Figure 2 summarizes our results. In Figure 2a, we depict the average of the maximum neuron correlations across models [b-e], the average of the baseline correlations, and the excess correlation i.e., the left term, the right term, and the difference in (3). While there is no principled threshold at which a neuron should be deemed universal, only 1253 out of the 98304 neurons in GPT2-medium-a have an excess correlation greater than 0.5. In Figure 14, we report the (complement) cumulative distribution of these correlation metrics to show how the number of universal neurons changes with threshold.

To understand if high (low) correlation in one model implies high (low) correlation in all the models, in Figure 2b we report $\max_m \max_{j \in N(x)} \rho_{i,j}^{a,m}$ compared to $\min_m \max_{j \in N(m)} \rho_{i,j}^{a,m}$ for every neuron $i \in N(a)$. Figure 2b suggests there is relatively little variation in the correlations, as the mean difference between the max-max and min-max correlation is 0.049 for all neurons and 0.105 for neurons with $\varrho > 0.5$. Another natural hypothesis is that neurons specialize into roles based on how deep they are within the network (as suggested by (Olah et al., 2020b; Elhage et al., 2022a)). In 2c, for each layer $l$ of model $a$, we compute the fraction of neurons in layer $l$ that have their most correlated neuron in layer $l'$ for all $l'$ in models [b-e]. Averaging across the different models, we observe significant *depth specialization*, suggesting that neurons do perform depth specific computations, which we explore further in § 4.3.

We repeat these experiments on GPT2-small and Pythia-160m displayed in Figures 12 and 13 respectively. A rather surprising finding is that while the percentage of universal neurons ($\varrho_i > 0.5$) within GPT2-medium and Pythia-160M are quite consistent (1.23% and 1.26% respectively), the number in GPT2-small-a is far higher at 4.16%. We offer additional results and speculations in § C.3.

## 4.2 Properties of Universal Neurons

We now seek to understand whether there are statistical proprieties associated with whether a neuron is universal or not, defined as having an excess correlation $\varrho_i > 0.5$. For all neurons in GPT2-medium-a, GPT2-small-a, and Pythia-160m, we compute various summary statistics of their weights and activations. For activations, we compute the mean, skew, and kurtosis of the pre-activation distribution over 100 million tokens, as well as the fraction of activations greater than zero, termed activation sparsity. For weights, we record the input bias $\mathbf{b}_{\text{in}}$, the cosine similarity between the input and output weight $\cos(\mathbf{w}_{\text{in}}, \mathbf{w}_{\text{out}})$, the weight decay penalty $\|\mathbf{w}_{\text{in}}\|_2^2 + \|\mathbf{w}_{\text{out}}\|_2^2$, and the kurtosis of the neuron output weights with the unembedding $\text{kurt}(\cos(\mathbf{w}_{\text{out}}, \mathbf{w}_U))$ to measure the composition with the unembedding (Geva et al., 2022; Dar et al., 2022).

In Figure 3, we report these statistics for universal neurons as a percentile compared to all neurons within the same layer; we choose this normalization to enable comparison across different layers, models, and metrics (a breakdown per metric and layer for GPT2-medium-a is given in Figure 15). Our results show that universal neurons do stand out compared to non-universal neurons. Specifically, universal neurons typically have large

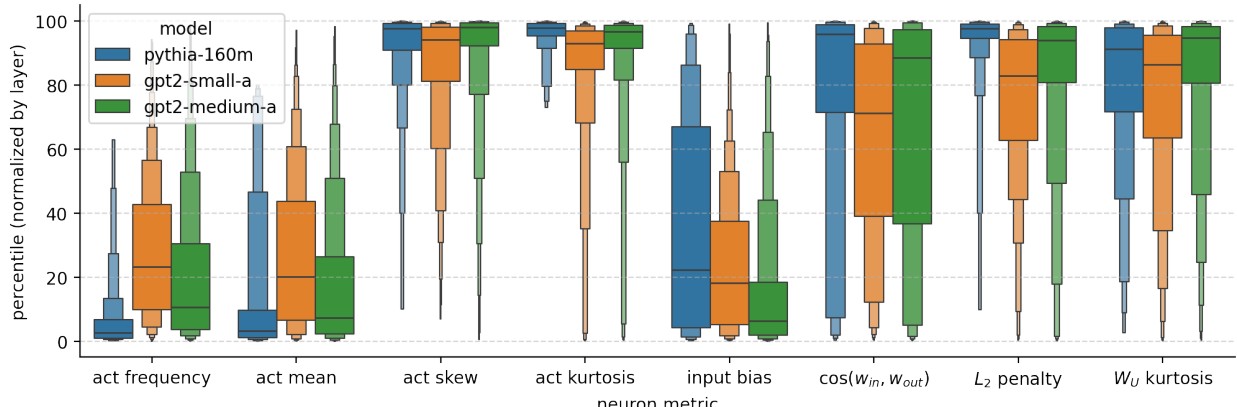

Figure 3: Properties of activations and weights of universal neurons for three different models, plotted as a percentile compared to neurons in the same layer.

weight norm (implying they are important because the model was trained with weight decay) and have a large negative input bias, resulting in a large negative pre-activation mean and hence lower activation frequency. Furthermore, universal neurons have very high pre-activation skew and kurtosis, implying they usually have negative activation, but occasionally have very positive activation, proprieties we would expect of monosemantic neurons (Olah et al., 2020b; Elhage et al., 2022b; Gurnee et al., 2023) which only activate when a specific feature is present in the input. In contrast, non-universal neurons usually have skew approximately 0 and kurtosis approximately 3, identical to a Gaussian distribution. We will discuss the meaning of high $\mathbf{W}_U$ kurtosis in § 5.1 and high $\cos(\mathbf{w}_{\text{in}}, \mathbf{w}_{\text{out}})$ in § C.

### 4.3 Universal Neuron Families

Motivated by the observation that universal neurons have distributional statistics suggestive of monosemanticity, we zoom-in on individual neurons with $\varrho > 0.5$ and attempt to group them into a partial taxonimization of neuron families (Olah et al., 2020a; Cammarata et al., 2021). After manually inspecting many such neurons, we developed several hundred automated tests to classify neurons using algorithmically generated labels derived from elements of the vocabulary (e.g., whether a token `is_all_caps` or `contains_digit`) and from the NLP package spaCy (Honnibal et al., 2020). Specifically, for each neuron with activation vector $\mathbf{v}$, and each test explanation which is a binary vector $\mathbf{y}$ over all tokens in the input, we compute the reduction in variance when conditioned on the explanation:

$$1 - \frac{(1 - \beta)\sigma^2(\mathbf{v}|\mathbf{y} = 0) + \beta\sigma^2(\mathbf{v}|\mathbf{y} = 1)}{\sigma^2(\mathbf{v})} \tag{4}$$

where $\beta$ is the fraction of positive labels and $\sigma^2(\cdot)$ is the variance of a vector or subset thereof. In words, Eq 4 measures the change in variance between the original activation distribution, and the weighted (by $\beta$) variance of the distribution slices where $y_i = 1$ and $y_i = 0$. As a useful intuition, this is the same metric used to decide how to split in a regression tree, where the goal is to find a split which most reduces the variance in the prediction target. Below, we qualitatively describe the most common families, and find our results replicate many findings previously documented in the literature.

**Unigram Neurons** The most common type of neuron we found were *unigram* neurons, which simply activate approximately if and only if the current token is a particular word or part of a word. These neurons often have many near duplicate neurons activating for the same unigram (Figure 16) and appear predominately in the first two layers (Figure 17). We breakdown activations of neurons responding to alphabetical unigrams based on the unigram's position in a word, as common words often have four tokenizations, and find that duplicate neurons can respond differently to unigram variations (Figures 4a and 16). Such neurons illustrate

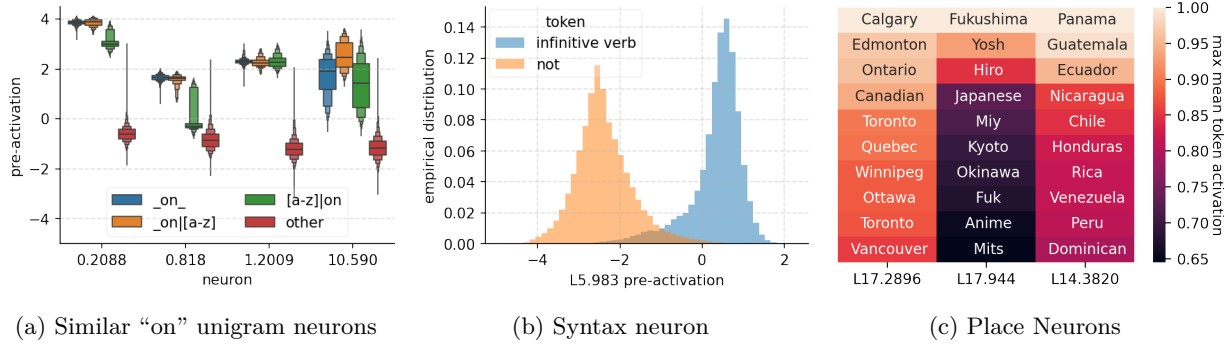

(a) Similar "on" unigram neurons    (b) Syntax neuron    (c) Place Neurons

Figure 4: Additional examples of universal neuron families in GPT2-medium.

that the token (un)embeddings may not contain all of the relevant token-level information, and that the model uses neurons to create an "extended" embedding of higher capacity.

**Alphabet Neurons**   A particularly fun subclass of unigram neurons are *alphabet* neurons (Figure 1a), which activate most strongly on tokens corresponding to an individual letter, and secondarily on tokens which begin with the respective letter. For 18 of 26 English letters there exist alphabet neurons with $\varrho > 0.5$ (Figure 18), with some letters also having several near duplicate neurons.

**Previous Token Neurons**   After finding an example of one neuron which seemed to activate purely as a function of the *previous* token (e.g., if it contains a comma; Figure 1b), we decided to rerun our unigram tests with the labels shifted by one—that is, with the label given by the previous token. These tests surfaced many more previous token neurons occurring most often in layers 4-6 (see Figure 19 for an additional 25 universal previous token neurons). Such neurons illustrate the many potentially redundant paths of computations that can occur which complicates ablation based interpretability studies.

**Position Neurons**   Inspired by the recent work of (Voita et al., 2023), we also run evaluations for *position neurons*, neurons which activate as a function of absolute position rather than token or context (Figure 1c). We follow the procedure of (Voita et al., 2023) (who run their experiments on OPT models with ReLU activation (Zhang et al., 2022)) by computing the mutual information between activation and context position, and find similar results, with neurons that have a variety of positional patterns concentrated in layers 0-2 (see Figure 20 for 20 more neurons). Similar to the unigram neurons, the presence of these neurons is potentially unexpected given their outputs could be learned directly by the positional embedding at the beginning of the model with less variance in activation.

**Syntax Neurons**   Using the NLP package spaCy (Honnibal et al., 2020), we label our input data with part-of-speech, dependency role, and morphological data. We find many individual neurons that selectively activate for basic linguistic features like negation, plurals, and verb forms (Figure 4b) which are not concentrated to any part of the network and resemble past findings on linguistic properties (Dalvi et al., 2019; Durrani et al., 2022). Figure 21 includes 25 more examples.

**Semantic Neurons**   Finally, we found a large number of neurons which activate for semantic features corresponding to coherent topics (Lim and Lauw, 2023), concepts (Elhage et al., 2022a), or contexts (Gurnee et al., 2023). Such features are naturally much harder to algorithmically supervise. We use the subdistribution label from the Pile dataset (Gao et al., 2020) and manually labeled topics from an SVD based topic model as a best attempt, but this leaves many interpretable neurons undiscovered and uncategorized. In Figure 4c, we show three regions neurons which activate most strongly on tokens corresponding to places in Canada, Japan, or Latin America respectively. Figure 22 depicts 30 additional context neurons which activate on specific subdistributions, with many neurons which always activate for non-english text.

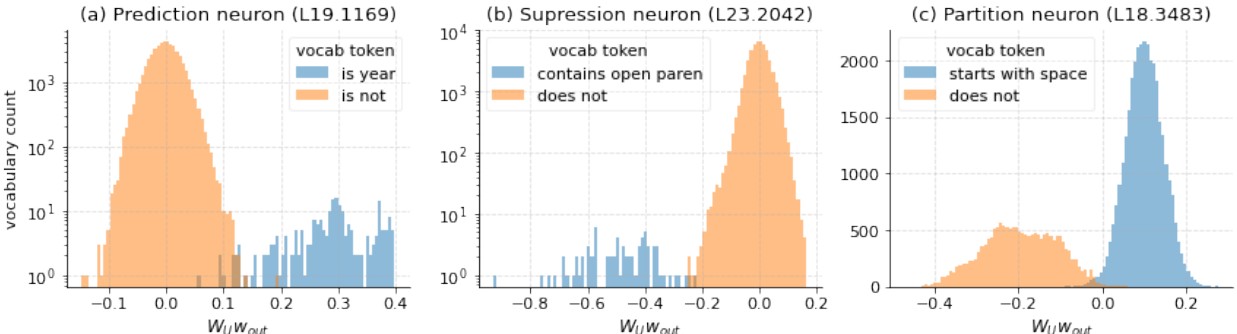

Figure 5: Example prediction neurons in GPT2-medium-a. Depicts the distribution of logit effects on the output vocabulary ($\mathbf{W}_U \mathbf{w}_{\text{out}}$) split by token property for 3 different neurons. (a) Prediction neuron increasing logits of integer tokens between 1700 and 2050 (i.e. years; high kurtosis), (b) Suppression neuron decreasing logits for tokens containing an open parenthesis (high kurtosis and negative skew), and (c) Partition neuron boosting tokens beginning with a space and suppressing tokens which do not (high variance).

## 5  Universal Functional Roles of Neurons

While the previous discussion was primarily focused on analyzing the *activations* of neurons, and by extension the features they represent, this section is dedicated to studying the *weights* of neurons to better understand their downstream effects. The neurons in this section are examples of *action* mechanisms (Anthropic, 2023)—model components that are better thought of as implementing an action rather than purely extracting or representing a feature, analogous to motor neurons in neuroscience.

### 5.1  Prediction Neurons

A simple but effective method to understand weights is through logit attribution techniques (Nostalgebraist, 2020; Geva et al., 2022; Dar et al., 2022). Because the final residual stream is the sum of all previous layers, we can approximate a neuron's effect on the final prediction logits by simply computing the product between the unembedding matrix and a neuron output weight $\mathbf{W}_U \mathbf{w}_{\text{out}}$ and hence interpret the neuron based on how it promotes concepts in the vocabulary space (Geva et al., 2022).

When we apply our automated tests from § 4.3 on $\mathbf{W}_U \mathbf{w}_{\text{out}}$ rather than the activations for our universal neurons, we found several general patterns (Figure 5), many individual neurons with extremely clear interpretations (Figure 24), and clusters of neurons which all affect the same tokens (Figure 25). Specifically, we find many examples of *prediction* neurons that positively increase the predicted probability of a coherent set of tokens while leaving most others approximately unchanged (Fig 5a); *suppression* neurons that are similar, except decrease the probability of a group of related tokens (Fig 5b); and *partition* neurons that partition the vocabulary into two groups, increasing the probability of one while decreasing the probability of the other (Fig 5c). The prediction, suppression, and partition motifs can be automatically detected by studying the moments of the distribution of vocabulary effects given by $\mathbf{W}_U \mathbf{w}_{\text{out}}$. In particular, both prediction and suppression neurons will have high kurtosis (the fourth moment—a measure of how much mass is in the tails of a distribution), but prediction neurons will have positive skew and suppression neurons will have negative skew. Partition neurons will shift the probability of most tokens and have high variance in overall logit effect. From this, we see almost all universal neurons ($\varrho > 0.5$) in later layers are one of these prediction neuron variants (Figure 15).

To better understand the number and location of these prediction neurons, we compute the moment metrics of $\cos(\mathbf{W}_U, \mathbf{w}_{\text{out}})$ for all neurons in all five GPT2-medium models, and show how these statistics vary over model depth in Figure 6. We find a striking pattern which is quite consistent across the different seeds: after about the halfway point in the model, prediction neurons become increasingly prevalent until the very end of the network where there is a sudden shift towards suppression neurons. To ensure this is not just an artifact

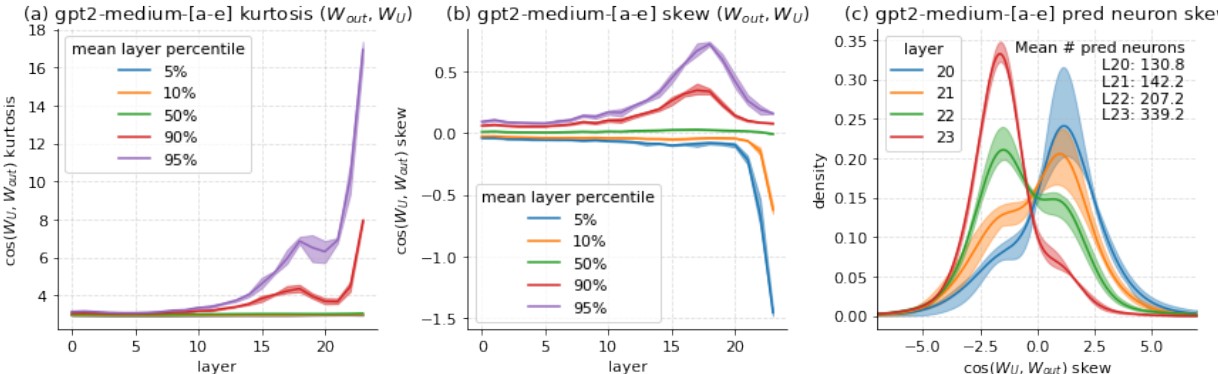

Figure 6: Summary statistics of cosine similarity between neuron output weights ($\mathbf{W}_{\text{out}}$) and token unembedding ($\mathbf{W}_U$) for GPT2-medium-[a-e]. (a,b) Percentiles of kurtosis and skew by layer averaged over [a-e]. (c) Distribution of skews for neurons with kurtosis greater than 10 in last four layers. Shaded area denotes range across all five models.

of the tied embeddings ($\mathbf{W}_E = \mathbf{W}_U^T$) in the GPT2 models, we also run this analysis on five Pythia models ranging from 410M to 6.9B parameters and find the results are largely the same (Figure 23).

We observed an interesting pattern when examining the activations of suppression neurons: they activate much more frequently when the next token actually belongs to the set of tokens they suppress the probability of predicting. In other words, neurons which *decrease* the probability that the next token is a year (e.g. "1970"), activate much more often when the next token is actually a year compared to not. We intuit that these suppression neurons fire when it is plausible but not certain that the next token is from the relevant set. Combined with the observation that there exist many suppression and prediction neurons for the same token class (Figure 25), we take this as evidence of an ensemble hypothesis where the model uses multiple neurons with some independent error that combine to form a more robust and calibrated estimate of whether the next token is in fact a year.

In addition to being a clean example of an action mechanism (Anthropic, 2023), these results are interesting as they refine a conjecture made by (Geva et al., 2022). Specifically, rather than "feed-forward layers build predictions by promoting concepts in the vocabulary space," we claim *late* feed-forward (MLP) layers build predictions by both promoting *and* suppressing concepts in the vocabulary space. Moreover, it suggests there are different stages in the iterative inference pipeline (Belrose et al., 2023; Jastrzębski et al., 2017), where first affirmative predictions are made, and then the distribution is sharpened or made more calibrated by suppression neurons at the very end. The existence of suppression neurons also sheds light on recent observations of individual neurons (Bills et al., 2023) and MLP layers (McGrath et al., 2023) suppressing the maximum likelihood token and being a mechanism for self-repair.

## 5.2 Entropy Neurons

Because models are trained with weight decay ($\ell_2$ regularization) we hypothesized that neurons with large weight norms would be more interesting or important because they come at a higher cost. While most turned out to be relatively uninteresting (mostly neurons which activate for the beginning of sequence token), the 15[th] largest norm neuron in GP2-medium-a (L23.945) had an especially interesting property: it had the lowest variance logit effect $\mathbf{W}_U\mathbf{w}_{\text{out}}$ of any neuron in the model; i.e., it only has a tiny effect on the logits. To understand why a final layer neuron, which can only affect the final logit distribution, has high weight norm while performing an approximate no-op on the logits, recall the final decoding formula for the probability of the next token given a final residual stream vector $\mathbf{x}$

$$p(\mathbf{y}|\mathbf{x}) = \text{Softmax}(\mathbf{W}_U \text{LayerNorm}(\mathbf{x})), \qquad \text{LayerNorm}(\mathbf{x}) = \frac{\mathbf{x} - \mathbb{E}[\mathbf{x}]}{\sqrt{\text{Var}[\mathbf{x}] + \epsilon}}. \tag{5}$$

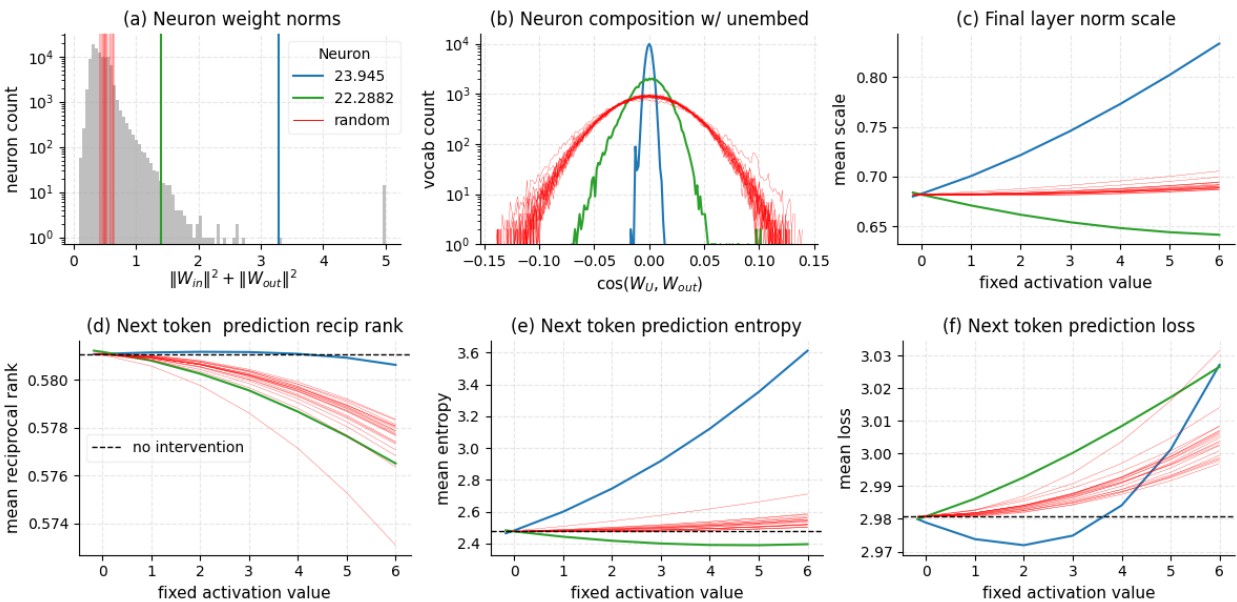

Figure 7: Summary of (anti-)entropy neurons in GPT2-medium-a compared to 20 random neurons from final two layers. Entropy neurons have high weight norm (a) with output weights mostly orthogonal to the unembedding matrix (b). Fixing the activation to larger values causes the final layer norm scale to increase dramatically (c) while leaving the ranking of the true next token prediction mostly unchanged (d). Increased layer norm scale squeezes the logit distribution, causing a large increase in the prediction entropy (e; or decrease for anti-entropy neuron) and an increase or decrease in the loss depending on the model's baseline level of under- or over-confidence (f). Legend applies to all subplots.

We hypothesize that the function of this neuron is to modulate the model's uncertainty over the next token by using the layer norm to squeeze the logit distribution, in a manner quite similar to manually increasing the temperature when performing inference. To support this hypothesis, we perform a causal intervention, fixing the neuron in question to a particular value and studying the effect compared to 20 random neurons from the last two layers that are not in the top decile of norm or in the bottom decile of logit variance (Figure 7). We find that intervening on this *entropy* neuron indeed causes the layer norm scale to increase dramatically (because of the large weight norm) while largely not affecting the relative ordering of the vocabulary (because of the low composition), having the effect of increasing overall entropy by dampening the post-layer norm component of $\mathbf{x}$ in the row space of $\mathbf{W}_U$.

Additionally, we observed a neuron (L22.2882) with $\cos(\mathbf{w}_{\text{out}}^{23.945}, \mathbf{w}_{\text{out}}^{22.2882}) = -0.886$ (i.e., a neuron that writes in the opposite direction forming an antipodal pair (Elhage et al., 2022b)) that also has high weight norm. Repeating the intervention experiment, we find this neuron *decreases* the layer norm scale and decreases the mean next token entropy, forming an anti-entropy neuron. These results suggest there may be one or more global uncertainty directions that the model maintains to modulate its overall confidence in its prediction. However, our experiments with fixed activation value do not necessarily imply the model uses these neurons to increase the entropy as a general uncertainty mechanism, and we did notice cases in which increasing the activation of the entropy neuron decreased entropy, suggesting the true mechanism may be more complicated.

We repeat these experiments on GPT2-small-a and find an even more dramatic antipodal pair of (anti-)entropy neurons in Figure 26. To our knowledge, this is the first documented mechanism for uncertainty quantification in language models and the second example of a mechanism involving layer norm (Brody et al., 2023).

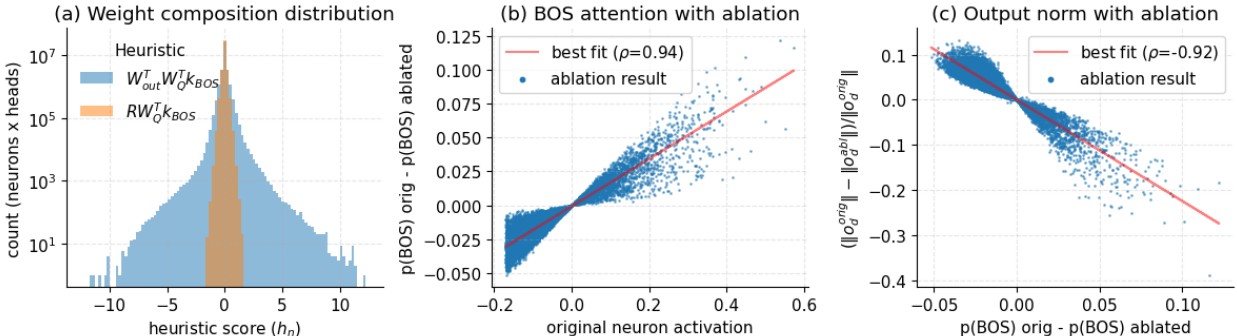

Figure 8: Summary of attention (de-)activation neuron results in GPT2-medium-a. (a) Distribution of heuristic score $h_n$ for every pair of neurons and heads compared to random neuron directions $\mathbf{R}$. (b;c) path ablations effect of neuron L4.3594 on head L5.H0: ablating positive activation reduces attention to BOS (b) causing the norm to increase (c).

## 5.3 Attention Deactivation Neurons

In autoregressive models, attention heads frequently place all of their attention on the beginning of sequence (BOS) token (Xiao et al., 2023). We hypothesise that the model uses the attention to the BOS token as a kind of (de-)activation for the head, where fully attending to BOS implies the head is deactivated and has minimal effect. Moreover, we hypothesize that there are individual neurons which control the extent to which heads attend to BOS.

Recall the output of an attention head $\mathbf{o}_d$ for a destination token $d$ from source tokens $s$ is given by

$$\mathbf{q}_d = \mathbf{W}_Q \mathbf{r}_d, \quad \mathbf{k}_s = \mathbf{W}_K \mathbf{r}_s, \quad \mathbf{S}_{ds} = \mathbf{q}_d^T \mathbf{k}_s, \quad \mathbf{A}_{ds} = \mathrm{softmax}_s(\frac{M(\mathbf{S}_{ds})}{\sqrt{d_h}}), \quad \mathbf{v}_s = \mathbf{W}_V \mathbf{r}_s, \quad \mathbf{o}_d = \mathbf{W}_O \sum_s \mathbf{A}_{ds} \mathbf{v}_s$$

where $\mathbf{r}_{s/d}$ is the residual stream at the source / destination token, $d_h$ is the bottleneck dimension of the head, and $M(\cdot)$ applies the causal attention mask to the attention scores. The calculation of the attention pattern $\mathbf{A}_{ds}$ via a softmax across the source positions means that the attention given to the source tokens by a given destination token sums to one.

Assuming the BOS token is always the first token in the context, the vector $\mathbf{W}_O \mathbf{v}_{BOS}$ is constant for all prompts and contains no semantic information. If it has a low norm, attending to BOS scales down the outputs of attending to other source positions while maintaining their relative attention because the attention scores must sum to one. If the BOS output norm is near zero, the head can effectively turn off by only attending to the BOS token. In practice, the median head in GPT-2-medium-a has a $\mathbf{W}_O \mathbf{v}_{BOS}$ with norm 19.4 times smaller than the average for other tokens.

We can identify neurons which may use this mechanism by a heuristic score $h_n = \mathbf{W}_{out}^T \mathbf{W}_Q^T \mathbf{k}_{BOS}$ for unit normalized $\mathbf{W}_{out}$. Positive scores suggest activation of the neuron will increase the attention placed on BOS, decreasing the output norm of the head, and the opposite for negative scores. Figure 8a shows the distribution of the scores for all heads in GPT2-medium-a compared to a unit normalized Gaussian matrix $\mathbf{R}$.

For a given neuron, we can measure the effect of activation on the attention to BOS and output norm of a given head by path ablation (Wang et al., 2022a) of the neuron at a particular destination token. Specifically, we can measure the difference in BOS attention and norm of the output of the head between the original run and a forward pass where the contribution of a neuron is deleted (i.e, zero path ablated) from the input of a particular head at the current token position. We perform this procedure over a random subset of tokens in the second half of the context to avoid spurious effects stemming from short contexts. Figure 8b and 8c depict the results of these path ablations for the highest scoring neuron in layer 4 for head 0 in attention layer 5. This is an example of an attention deactivation neuron—increasing the activation of the neuron

increases the attention to BOS reducing the output norm of the head $\|\mathbf{o}_d\|$. See Figure 27 for 5 additional examples of attention (de-)activating neurons.

## 6 Discussion and Conclusion

**Findings**   In this work, we explore the universality of individual neurons in GPT2 language models, and find that only about 1-5% of neurons pass a certain threshold of universality across models. We have shown that leveraging universality is an effective unsupervised approach to identify interpretable model components and important motifs. In particular, those few neurons which are universal are often interpretable, can be grouped into a smaller number of neuron families, and often develop with near duplicate neurons in the same model. Some universal neurons also have clear functional roles, like modulating the next token prediction entropy, controlling the output norm of an attention head, and predicting or suppressing elements of the vocabulary in the prediction. Moreover, these functional neurons often form antipodal pairs, potentially enabling collections of neurons to ensemble to improve robustness and calibration. These findings raise useful lessons and motifs for further interpretability research (§A.2).

**Limitations**   Compared to frontier LLMs, we study small models of only hundreds of million parameters and tens of thousands of neurons due to the expense of training multiple large scale language models from different random initializations. We also study a relatively narrow form of universality: neuron universality over random seeds within the same model family. Studying universality across different model families is made difficult by tokenization discrepancies, and studying models across larger sizes is difficult due to the expense of computing all pairwise neuron correlations over a sufficiently sized text corpus. Additionally, many of our interpretations rely on manual analysis or algorithmic supervision which restricts the scope and generality of our methods. Moreover, our narrow focus on a subset of individual elements of the neuron basis potentially obscures important details and ignores the vast majority of overall network computation.

**Future Work**   Each of these limitations suggest avenues for future work. Instead of studying the neuron basis, our experiments could be replicated on an overcomplete dictionary basis that is more likely to contain the true model features (Cunningham et al., 2023; Bricken et al., 2023). Motivated by the finding that the most correlated neurons occur in similar network depths, our experiments could be rerun on larger models where pairwise correlations are only computed between adjacent layers to improve scalability. Additionally, the interpretation of common units could be further automated using LLMs to provide explanations (Bills et al., 2023). Finally, by uncovering interpretable footholds within the internals of the network, our findings can form the basis of deeper investigations into how these components respond to stimulus or perturbation, develop over training (Quirke et al., 2023), and affect downstream components to further elucidate general motifs and specific circuits within language models.

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

# A    Additional Discussion

### A.1    Why expect universal neurons to be more monosemantic?

The intuition for the connection comes from Elhage et al. (2022b) which showed that important features (those that are constructed to have maximum effect on the loss) are the features which get their own dedicated neuron (making the neuron monosemantic). Since these GPT2 models are training on the same data, the important features should be similar across the networks. So if monosemantic neurons primarily represent important features, and the important features are shared (i.e., universal) across the networks, then we would expect the neurons which are universal across the networks to be representing the same set of important features.

Another line of argument is to consider the probability of a polysemantic neuron being universal. It is extremely unlikely that there would exist a pair of neurons which activates for the same $k \gg 0$ unrelated features, assuming unrelated features have roughly similar probabilities of being assigned to neurons. Moreover, this simple model suggests that as $k \to 1$ it is much more likely for there to exist a neuron which represents the same set of features.

### A.2    What are these observations useful for?

Our motivation here is primarily to improve our understanding of models for interpretability sake, rather than make models more performant. Specifically, we think interpretability is largely an immature field, without much grounding theory. Therefore, we think it's inherently valuable to gain a lot of "empirical surface area" on real networks to constrain the hypothesis space and develop the theory and practice of interpretability.

Example insights from our paper that might help future interpretability researchers

- Single neuron ablations can be misleading if there is an ensemble of similar neurons or there are neurons which consistently cancel each other out.

- The depth specialization results and the prediction followed by suppression neuron transition emphasize the sequential and residual nature or model processing.

- Our results on entropy neurons exemplify a concrete case where it would not be valid to linearize layer norm.

- Our results on attention deactivation neurons show how certain features might be difficult to interpret, because they are features which take internal actions rather than represent external inputs.

# B    Additional Empirical Details

All of our code and data is available at `https://github.com/wesg52/universal-neurons`.

Most of our plots in the main text (and therefore neuron indices) correspond to the HuggingFace model `stanford-crfm/arwen-gpt2-medium-x21` with our additional correlation experiments being conducted on `stanford-crfm/alias-gpt2-small-x21` and `EleutherAI/pythia-160m`.

## B.1    Weight Preprocessing

We employ several standard weight preprocessing techniques to simplify calculations (Nanda, 2022).

**Folding in Layer Norm**    Most layer norm implementations also include trainable parameters $\boldsymbol{\gamma} \in \mathbb{R}^n$ and $\mathbf{b} \in \mathbb{R}^n$

$$\text{LayerNorm}(\mathbf{x}) = \frac{\mathbf{x} - \mathbb{E}(\mathbf{x})}{\sqrt{\text{Var}(\mathbf{x})}} * \boldsymbol{\gamma} + \mathbf{b}. \tag{6}$$

To account for these, we can "fold" the layer norm parameters in to $W_{\text{in}}$ by observing that the layer norm parameters are equivalent to a linear layer, and then combine the adjacent linear layers. In particular, we can create effective weights

$$\mathbf{W}_{\text{eff}} = \mathbf{W}_{\text{in}} \ \mathbf{diag}(\boldsymbol{\gamma}) \qquad \mathbf{b}_{\text{eff}} = \mathbf{b}_{\text{in}} + \mathbf{W}_{\text{in}}\mathbf{b} \tag{7}$$

Finally, we can center the reading weights because the preceding layer norm projects out the all ones vector. Thus we can center the weights $\mathbf{W}_{\text{eff}}$ becomes

$$\mathbf{W}'_{\text{eff}}(i,:) = \mathbf{W}_{\text{eff}}(i,:) - \bar{\mathbf{W}}_{\text{eff}}(i,:)$$

**Writing Weight Centering**    Every time the model interacts with the residual stream it applies a LayerNorm first. Thus the components of $\mathbf{W}_{\text{out}}$ and $\mathbf{b}_{\text{out}}$ that lie along the all-ones direction of the residual stream have no effect on the model's calculation. So, we again mean-center $\mathbf{W}_{\text{out}}$ and $\mathbf{b}_{\text{out}}$ by subtracting the means of the columns of $\mathbf{W}_{\text{out}}$

$$\mathbf{W}'_{\text{out}}(:,i) = \mathbf{W}_{\text{out}}(:,i) - \bar{\mathbf{W}}_{\text{out}}(:,i)$$

**Unembed Centering**    Additionally, since softmax is translation invariant, we modify $\mathbf{W}_U$ into

$$\mathbf{W}'_{\text{U}}(:,i) = \mathbf{W}_{\text{U}}(:,i) - \mathbf{w}_i$$

For both of theses, see the transformer lens documentation for more details.

The purpose of all of these translations is to remove irrelevant components and other parameterization degrees of freedom so that cosine similarities and other weight computations have mean 0.

## B.2    Correlation Computations

We compute our correlations over a 100 million token subset of the Pile test set (Gao et al., 2020), tokenized to a context length of 512 tokens. We compute correlations over all tokens that are not padding, beginning-of-sequence, or new-line tokens.

**Efficient Computation**    Because storing neuron activations for two models over 100M tokens would be 36 petabytes of data, we require a streaming algorithm. To do so, observe that given a pair of neuron activations $\{(x_1, y_1), \ldots, (x_n, y_n)\}$ consisting of $n$ pairs, the correlation can be computed as

$$\rho_{xy} = \frac{\sum_{i=1}^n (x_i - \bar{x})(y_i - \bar{y})}{\sqrt{\sum_{i=1}^n (x_i - \bar{x})^2}\sqrt{\sum_{i=1}^n (y_i - \bar{y})^2}} = \frac{\sum_i x_i y_i - n\bar{x}\bar{y}}{\sqrt{\sum_i x_i^2 - n\bar{x}^2}\sqrt{\sum_i y_i^2 - n\bar{y}^2}}$$

where $\bar{x}, \bar{y}$ are the sample mean. Therefore, instead of saving all neuron activations, we can maintain four `n_neuron` dimensional vectors and one `n_neuron` × `n_neuron` matrix corresponding to the running neuron activation means in model A and model B, a running sum of each neurons squared activation, and a running sum of pairwise products. At the end of the dataset, we perform the appropriate arithmetic to combine the results into pairwise correlations for all models.

### B.3   Model Hyperparameters

| Property | GPT-2 Small | GPT-2 Medium | Pythia 160M |
|---|---|---|---|
| layers | 12 | 24 | 12 |
| heads | 12 | 16 | 12 |
| $d_{\mathrm{model}}$ | 768 | 1024 | 768 |
| $d_{\mathrm{vocab}}$ | 50257 | 50257 | 50304 |
| $d_{\mathrm{MLP}}$ | 3072 | 4096 | 3072 |
| parameters | 160M | 410M | 160M |
| context | 1024 | 1024 | 2048 |
| activation function | gelu_new | gelu_new | gelu |
| pos embeddings | absolute | absolute | RoPE |
| rotary percentage | N/A | N/A | 25 |
| precision | Float-32 | Float-32 | Float-16 |
| dataset | OpenwebText | OpenwebText | Pile |
| $p_{\mathrm{dropout}}$ | 0.1 | 0.1 | 0 |

Table 1: Hyperparameters of models

## C   Additional Mysteries

We conclude our investigation by commenting on several miscellaneous results that we think are worth reporting but that we do not fully understand.

### C.1   Cosine and Activation Frequency

An unexpectedly strong relationship we observed is the correlation between activation frequency of a neuron and the cosine similarity between its input and output weight vectors $\cos(\mathbf{w}_{\mathrm{in}}, \mathbf{w}_{\mathrm{out}})$ as shown in Figure 9. Almost all neurons with a very high activation frequency have input and output weights in almost opposite directions. These neurons are predominantly in the first quarter of network depth and have small excess correlation, i.e., they are not universal as measured by activation. We also find it noteworthy that there appears to be an approximate ceiling and floor on the cosine similarity of approximately $\pm 0.8$.

### C.2   Duplication and Universality

While neuron redundancy has been observed in models before (Casper et al., 2021; Dalvi et al., 2020) and large models can be effectively pruned (Xia et al., 2023), we were surprised by the number of seemingly duplicate universal neurons we observed (e.g., Figure 16 or the 105 BOS neurons we observed). Naively, this is surprising, as it seems wasteful to dedicate multiple neurons to the same feature. Larger models have more capacity and are empirically much more effective so why have redundant neurons when you could instead have one neuron with twice the output weight norm?

A few potential explanations are (1) these models were trained with weight decay, creating an incentive to spread out the computation. (2) Dropout—however, in these models dropout is applied to the output of the MLP layer, rather than the MLP activations themselves. (3) These neurons are vestigial remnants that were useful earlier in training (Quirke et al., 2023), but are potentially stuck in a local minima and are no longer useful. (4) The duplicated neurons are only activating the same on common features, but are polysemantic

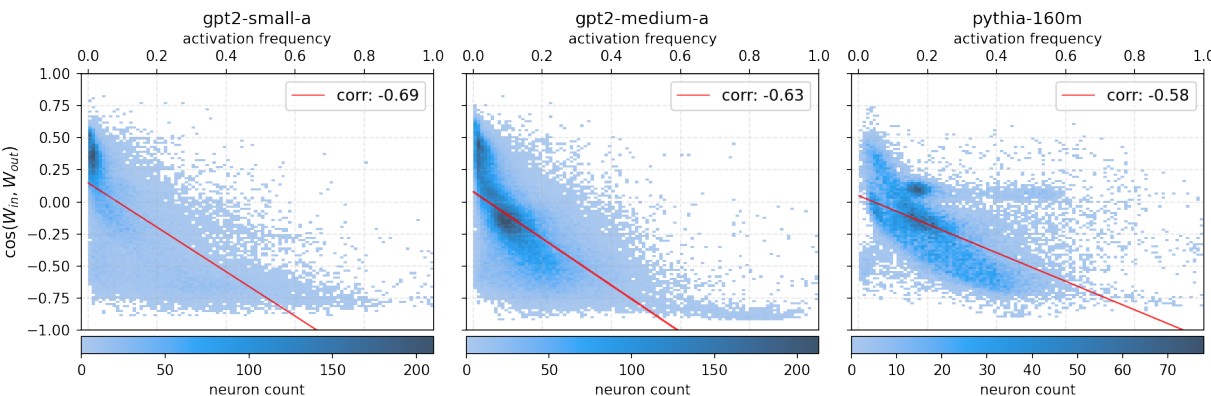

Figure 9: Activation frequency of neuron (fraction of activation values greater than zero) versus cosine similarity of neuron input and output weights for GPT2-small-a (left), GPT2-medium-a (center), and Pythia-160M (right).

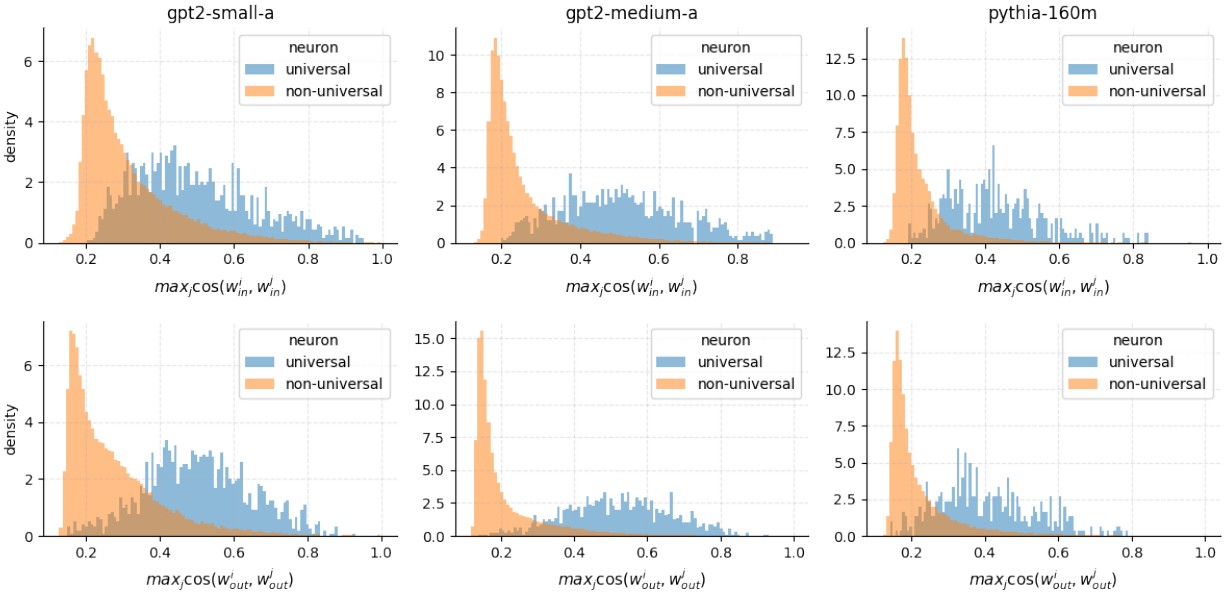

Figure 10: Distribution of cosine similarities of most similar neurons measured by input weights (top) and output weights (bottom) for GPT2-small-a (left), GPT2-medium-a (middle), and Pythia-160M (right) colored by universality ($\varrho > 0.5$).

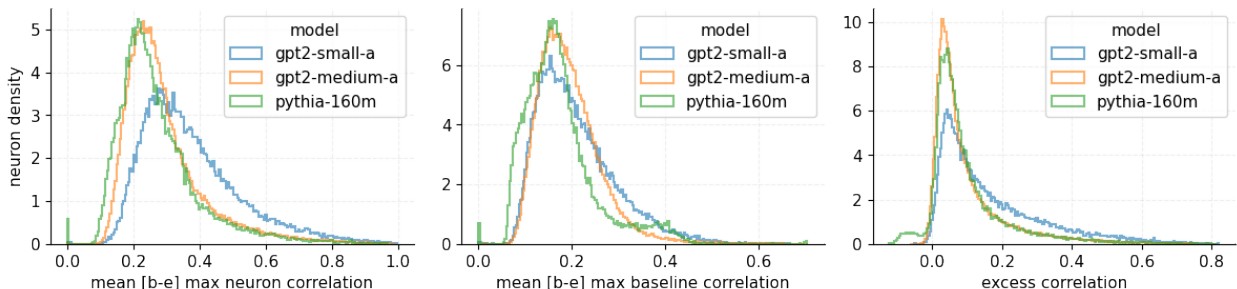

Figure 11: Empirical distribution of max neuron correlation averaged across models (left), max baseline correlation averaged across models (middle), and the difference denoted as the excess correlation (right).

with different sets of rarer features. (5) Ensembling, where each neuron computes the same feature but with some independent error, and together form an ensemble with lower average error.

By measuring redundancy in terms of similarity in weights (Figure 10), we find very few neurons which are literal duplicates, providing more evidence for (4) and (5). Based on the much higher level of similarity for universal neurons, it is possible this effect is relatively small in general.

### C.3   Scale and Universality

As mentioned in § 4, GPT2-medium and Pythia-160M have a consistent number of universal neurons (1.23% and 1.26% respectively), while GPT2-small-a has many more 4.16%. In Figure 11 we show the distribution of max, baseline, and excess correlations for all models, where we see that GPT2-medium and Pythia-160M have almost identical distributions while GPT2-small is an outlier. GPT2-small also has correspondingly greater weight redundancy as shown in Figure 10. One explanation for this is the number of universal neurons decreases in larger models. This is potentially implied by results from (Bills et al., 2023) who observe larger models have fewer neurons which admit high quality natural language interpretations. However, without additional experiments on larger models trained from random seeds, this remains an open question.

## D   Additional Results

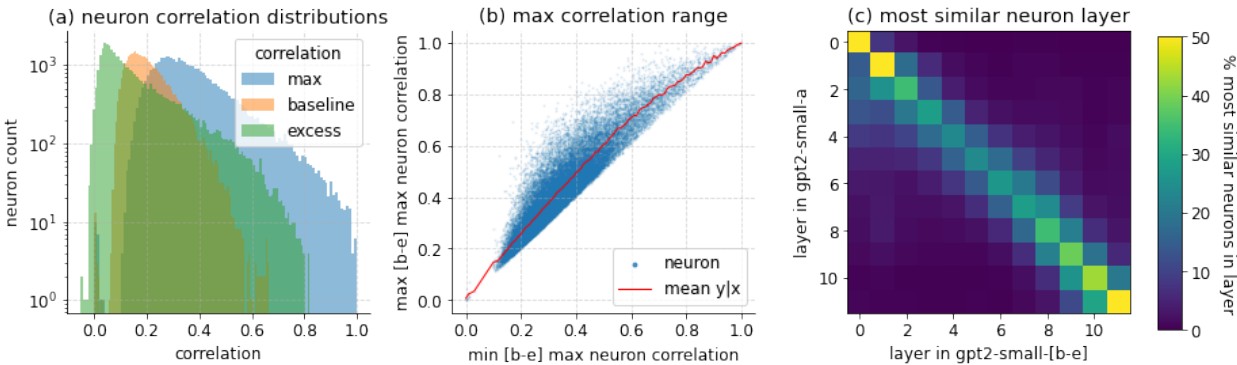

Figure 12: Summary of neuron correlation experiments in GPT2-small-a. (a) Distribution of the mean (over models b-e) max (over neurons) correlation, the mean baseline correlation, and the difference (excess). (b) The max (over models) max (over neurons) correlation compared to the min (over models) max (over neuron) correlation for each neuron. (c) Percentage of layer pairs with most similar neuron pairs.

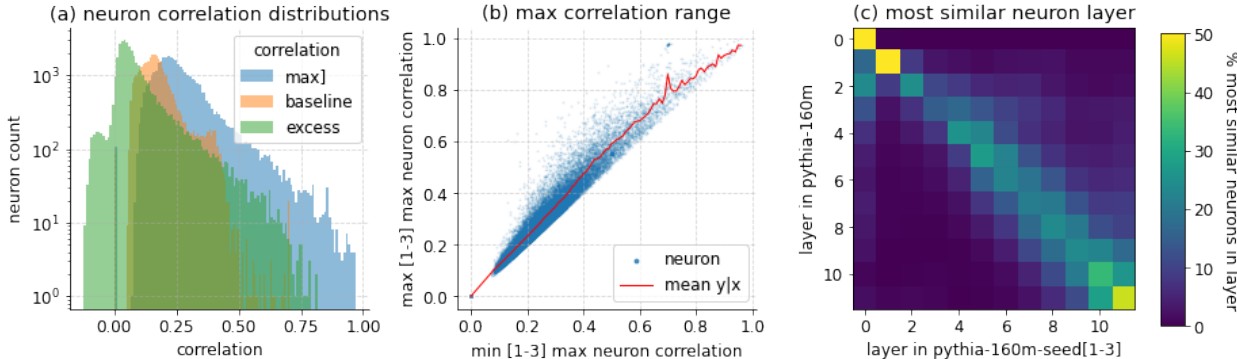

Figure 13: Summary of neuron correlation experiments in Pythia-160m. (a) Distribution of the mean (over models b-e) max (over neurons) correlation, the mean baseline correlation, and the difference (excess). (b) The max (over models) max (over neurons) correlation compared to the min (over models) max (over neuron) correlation for each neuron. (c) Percentage of layer pairs with most similar neuron pairs.

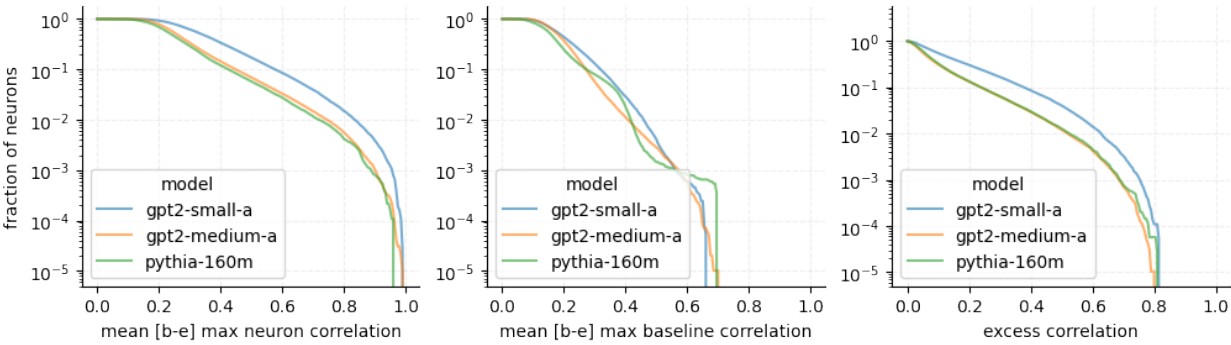

Figure 14: Complement cumulative distribution function for correlation metrics across all model families.

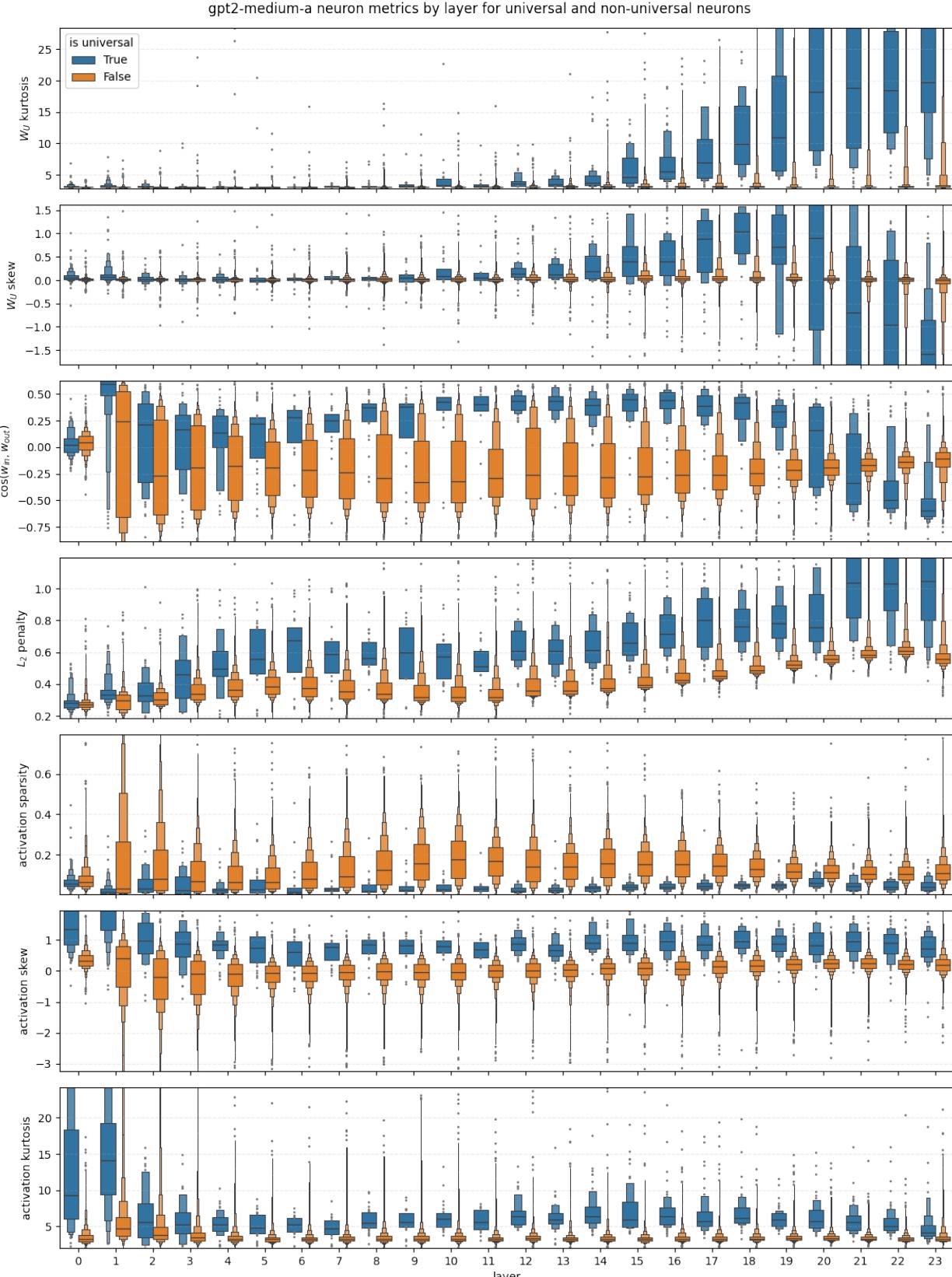

Figure 15: Distribution of neuron metrics for universal and non-universal neurons in GPT2-medium-a by layer. From top to bottom: the kurtosis of $\cos(\mathbf{W}_U, \mathbf{w}_{out})$, the skew of $\cos(\mathbf{W}_U, \mathbf{w}_{out})$, cosine similarity between input and output weight $\cos(\mathbf{w}_{in}, \mathbf{w}_{out})$, weight decay penalty $\|\mathbf{w}_{in}\|_2^2 + \|\mathbf{w}_{out}\|_2^2$, activation frequency (percentage of activations greater than 0), the pre-activation skew, and the pre-activation kurtosis.

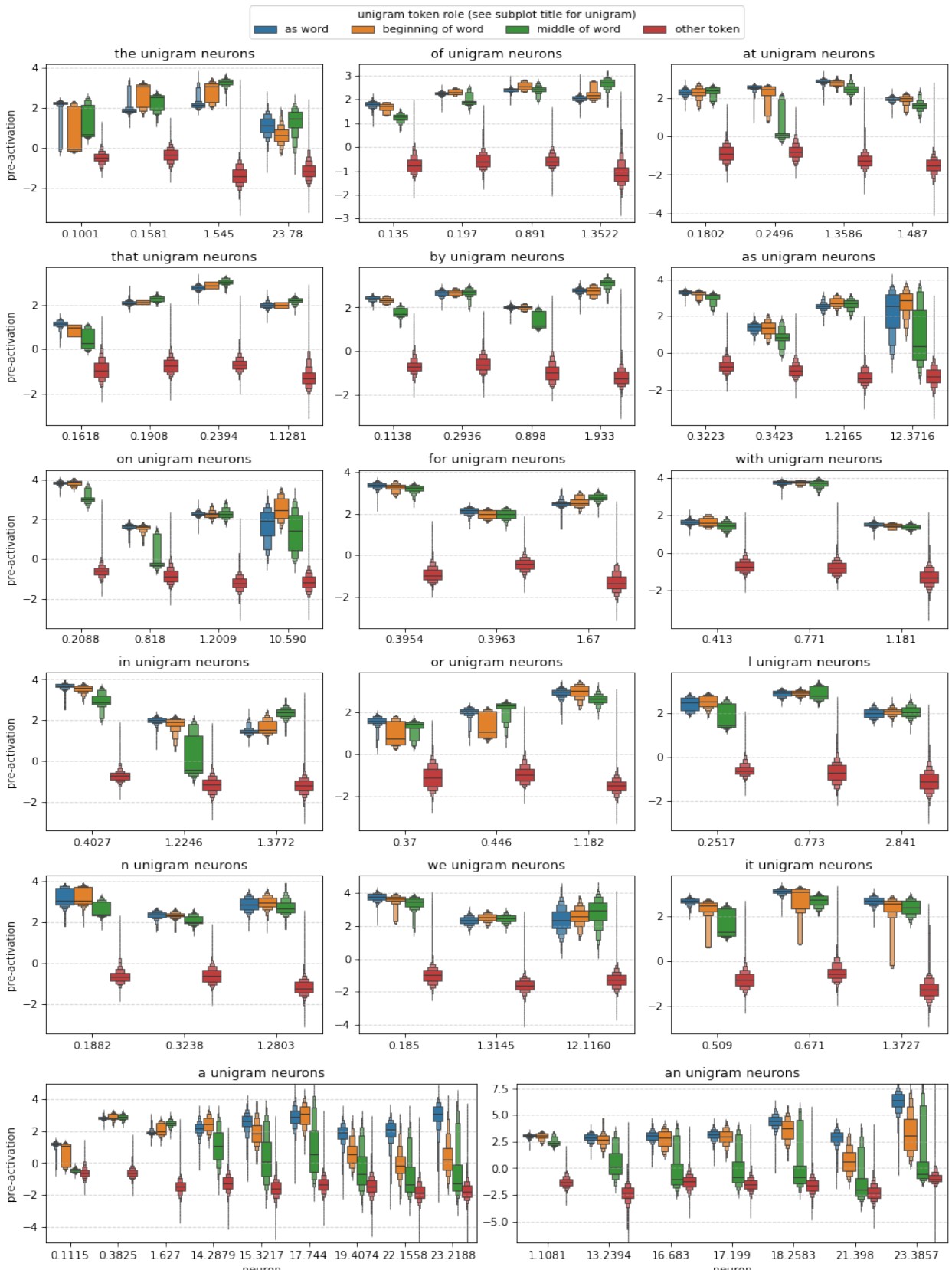

Figure 16: Duplicate unigram neurons in GPT2-medium-a. Each subplot depicts several neurons which activate on a particular token, broken down by whether this token exists as a standalone word, is the first token in a multi-token word, or is a non-first token in a multi-token word, versus all other tokens (e.g., "an," "an|agram," "Gig|an|tism").

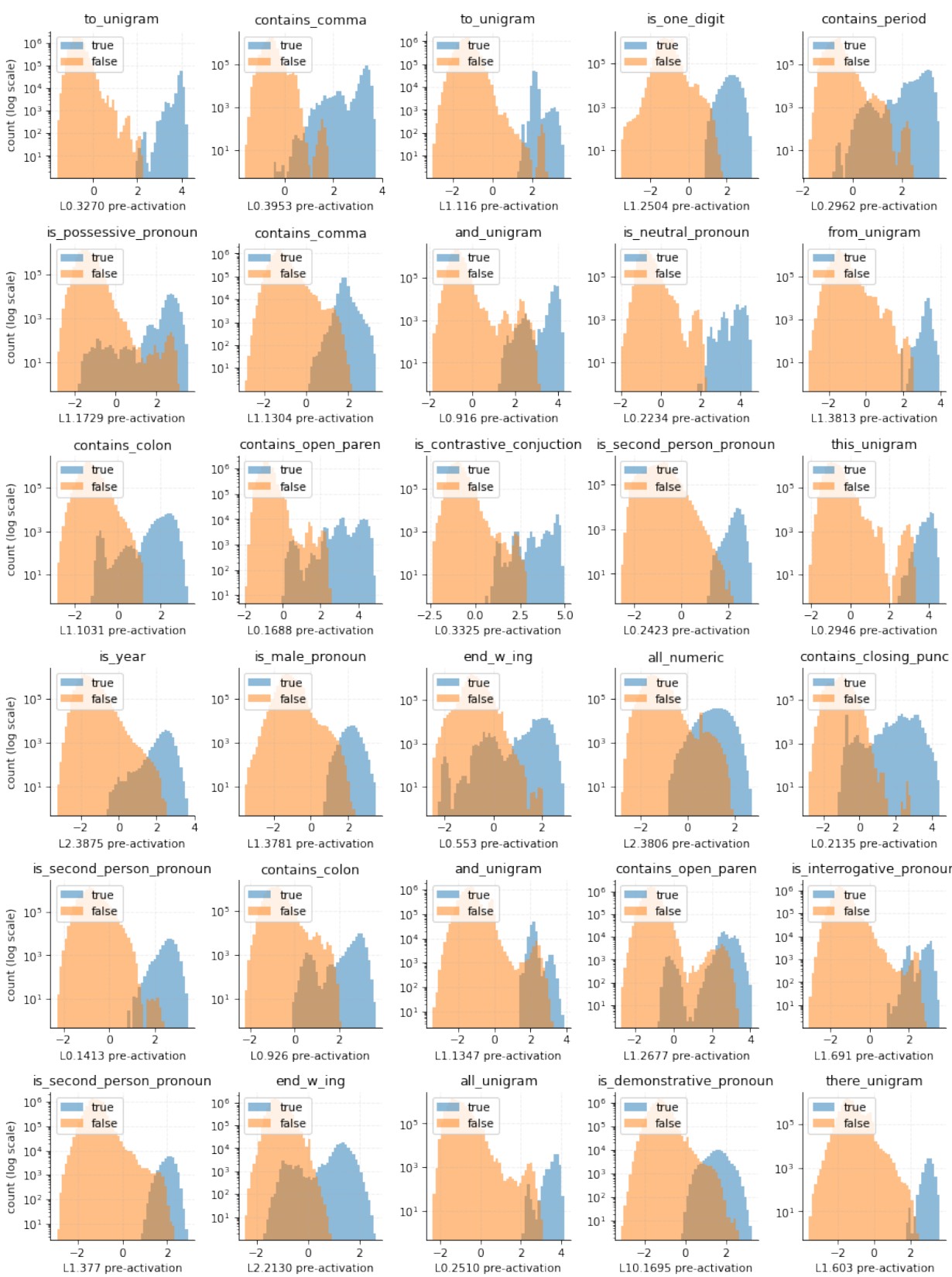

Figure 17: Universal unigram neurons in GPT2-medium-a.

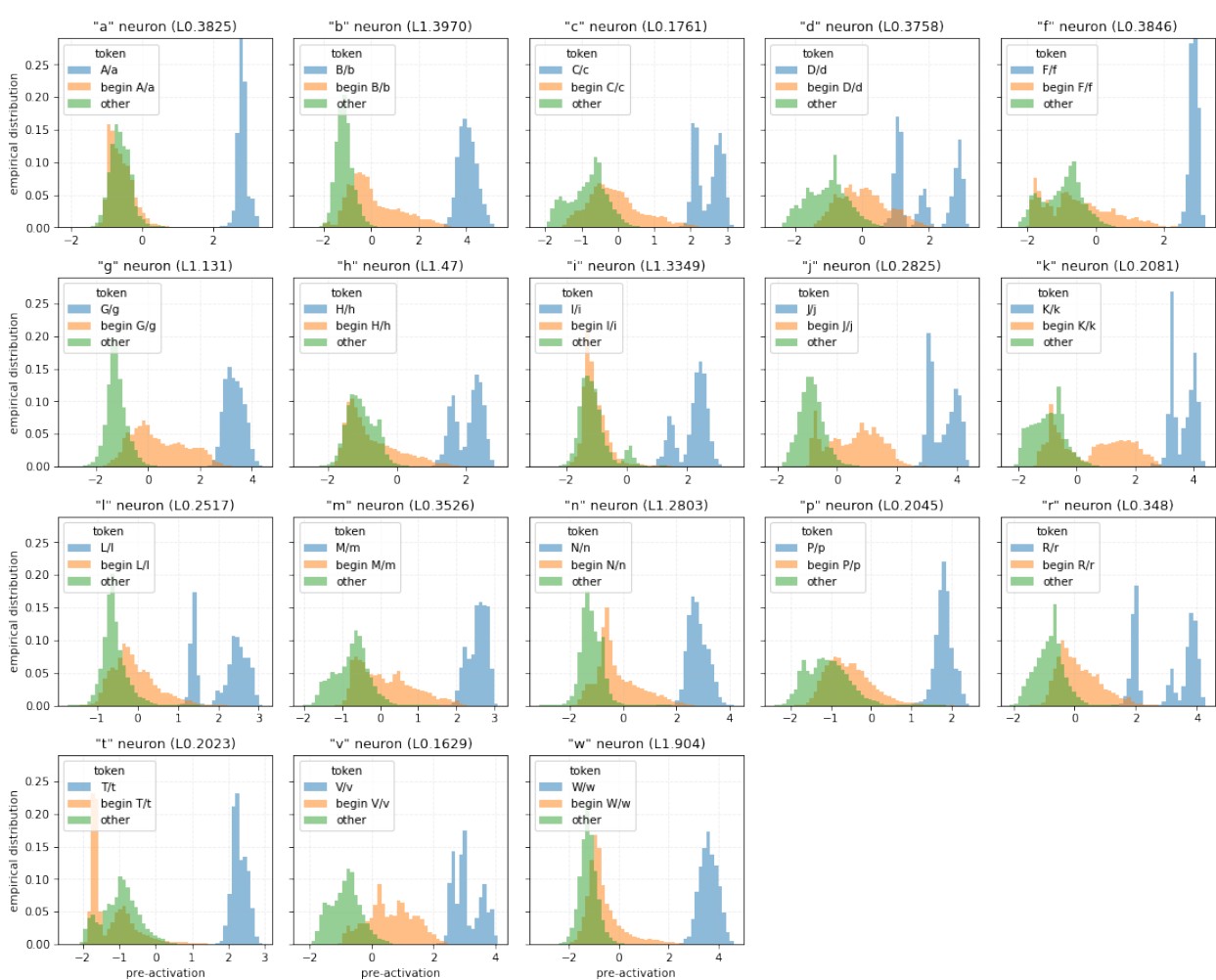

Figure 18: Universal alphabet neurons in GPT2-medium-a.

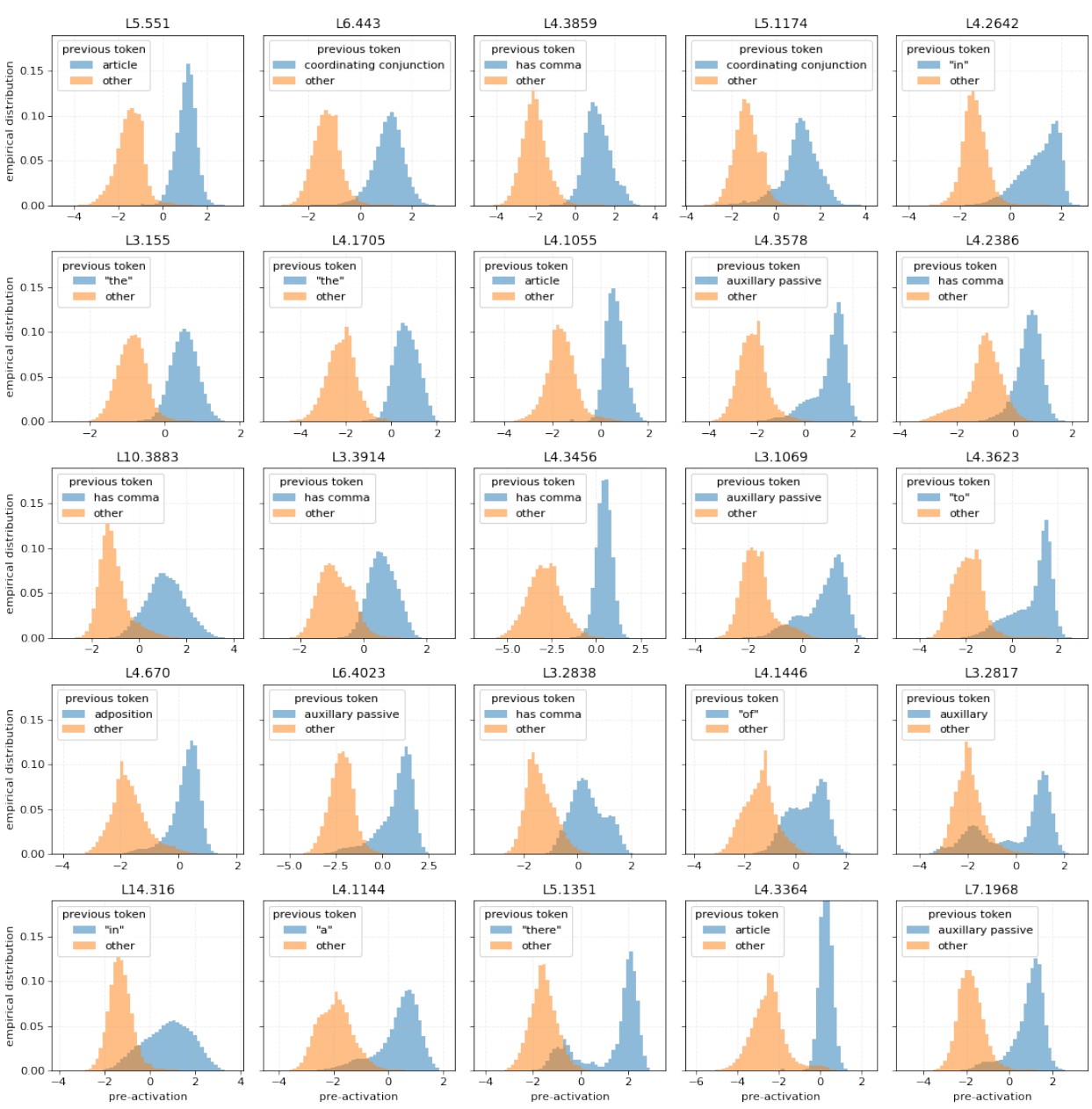

Figure 19: Universal previous token neurons in GPT2-medium-a.

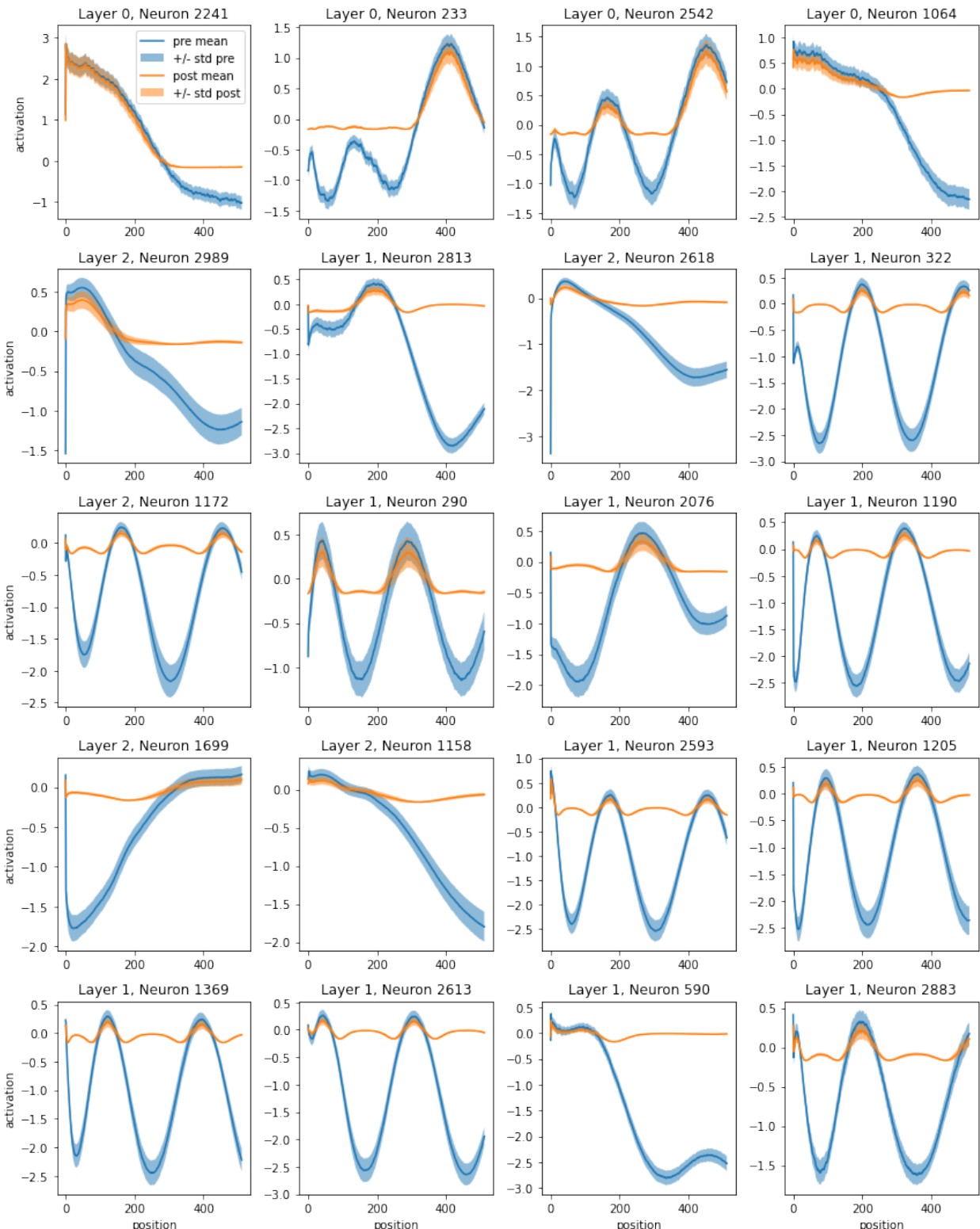

Figure 20: Universal position neurons in GPT2-small-a.

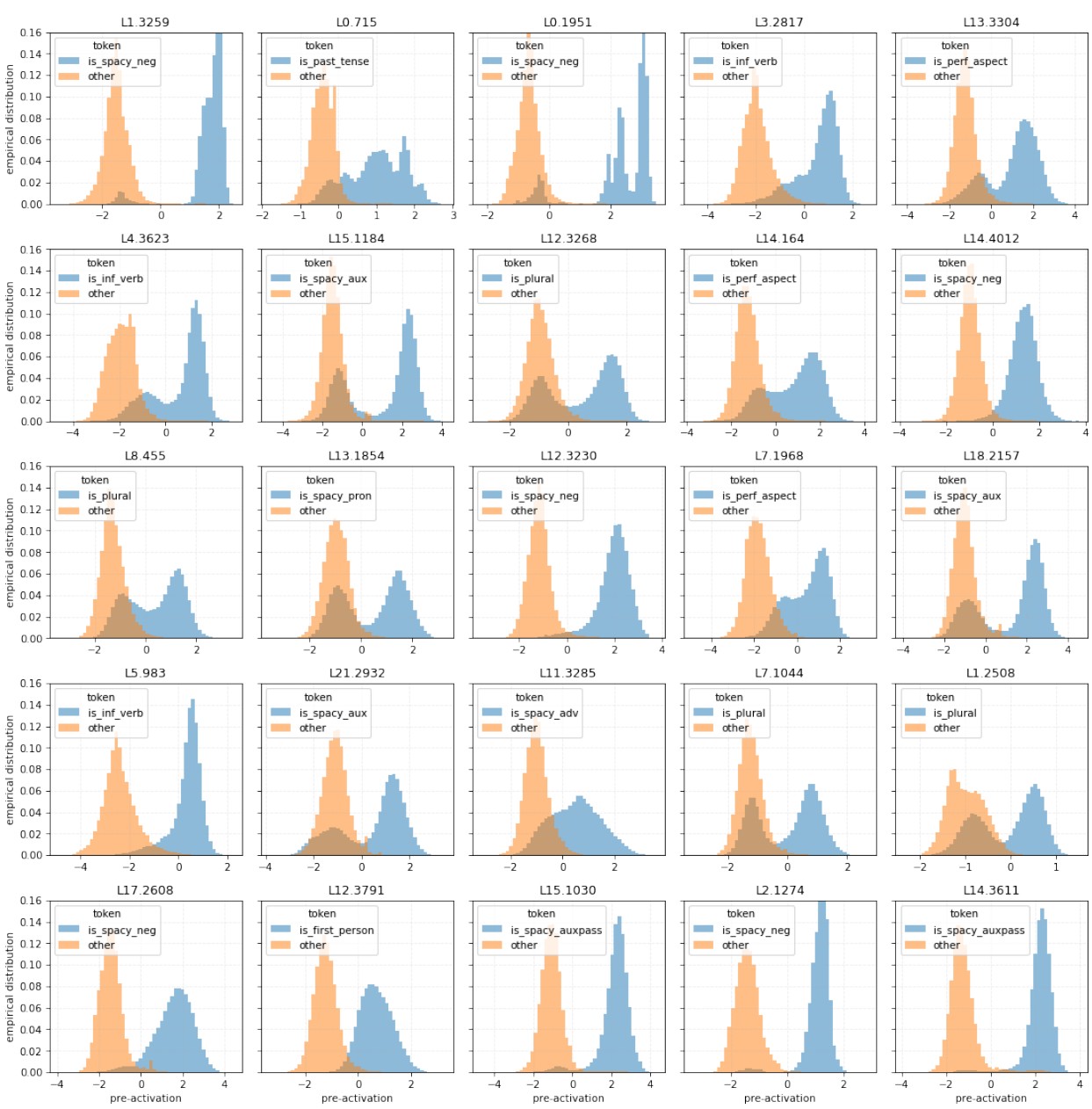

Figure 21: Universal syntax neurons in GPT2-medium-a.

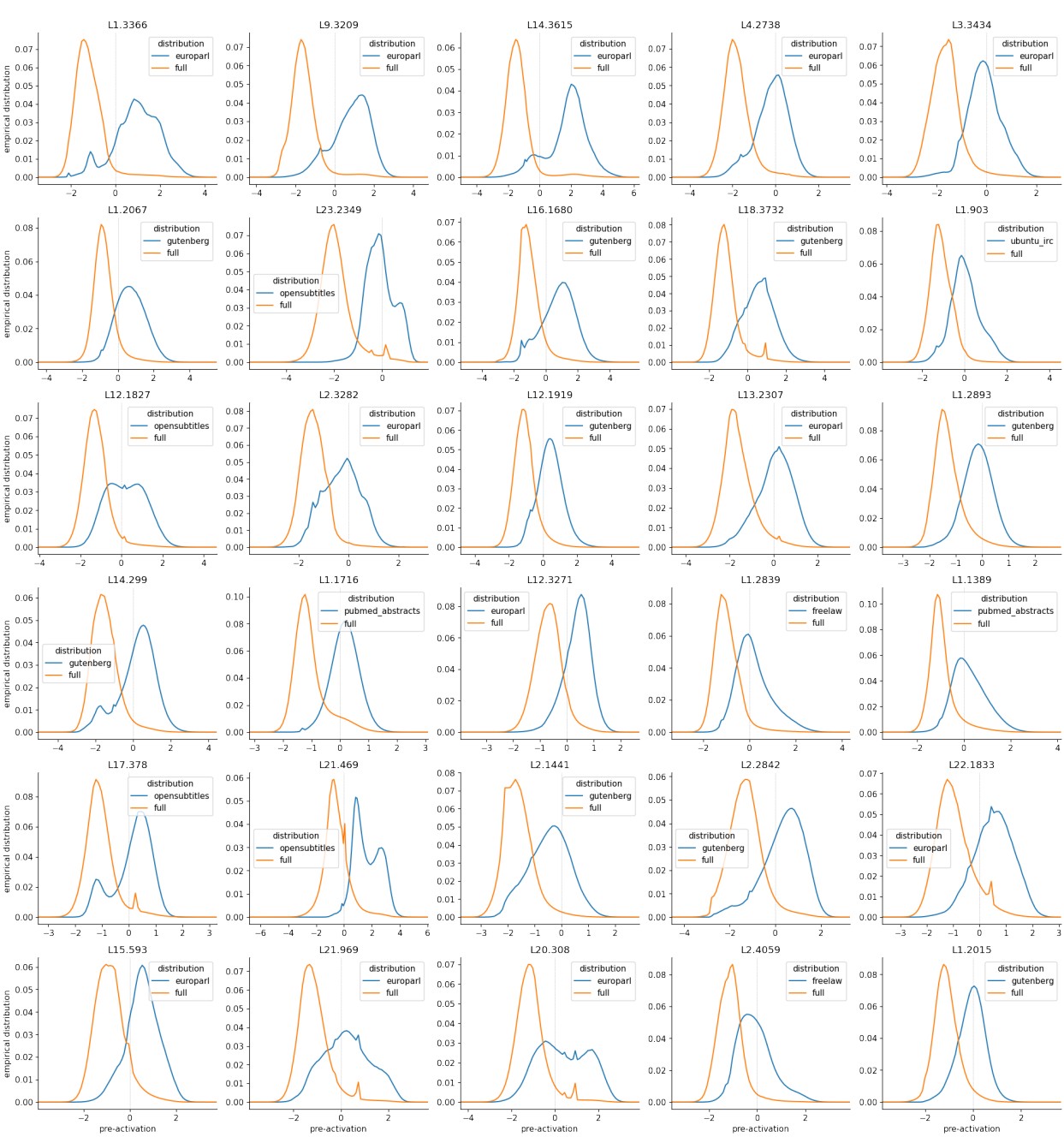

Figure 22: Universal context neurons in GPT2-medium-a.

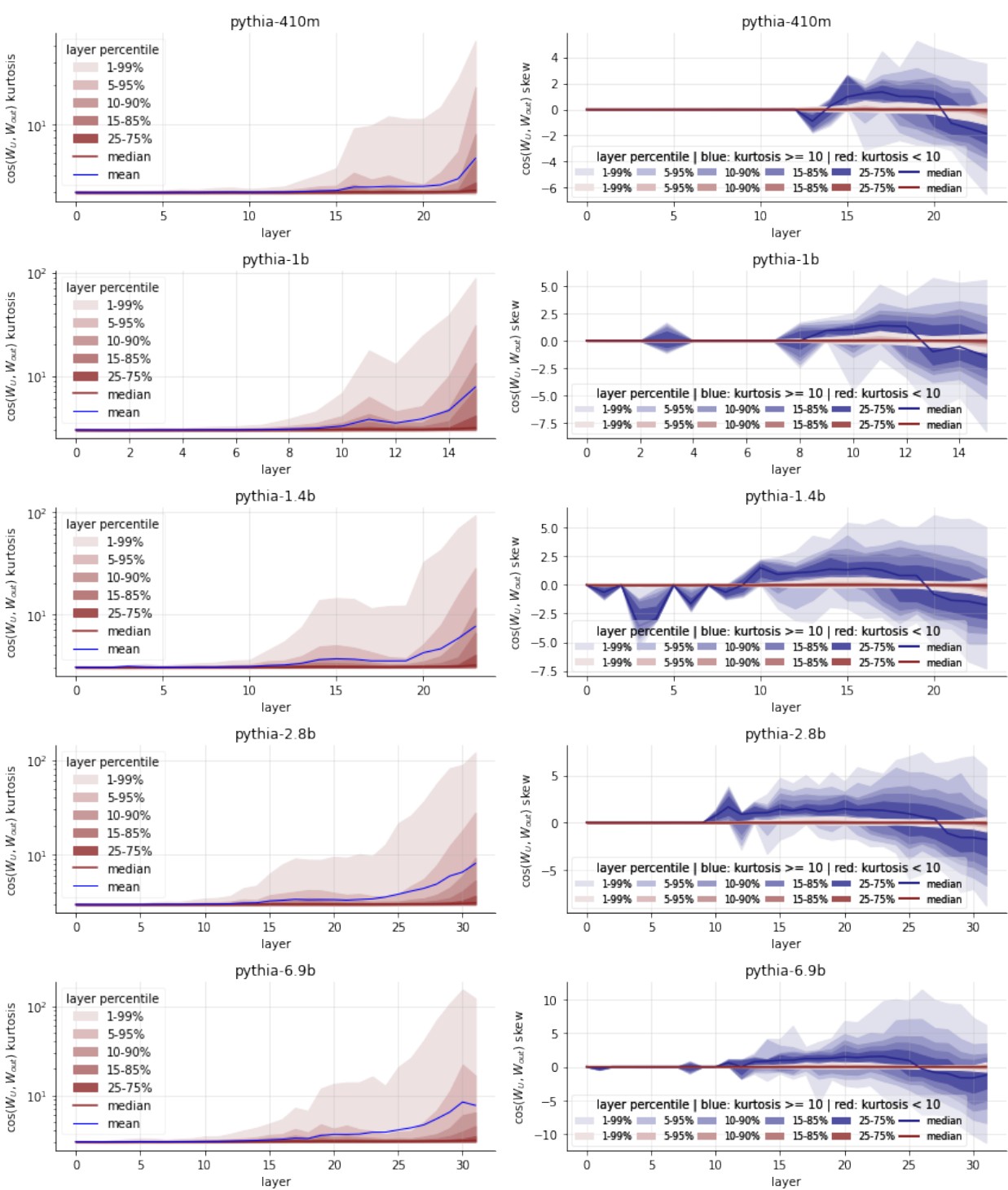

Figure 23: Distribution of vocabulary composition statistics for five different Pythia models measured over layers. Left shows percentiles of $\cos(\mathbf{W}_U, \mathbf{W}_{out})$ kurtosis. Right shows percentiles of $\cos(\mathbf{W}_U, \mathbf{W}_{out})$ skew broken down by whether neuron has $\cos(\mathbf{W}_U, \mathbf{W}_{out})$ kurtosis greater than or less than 10.

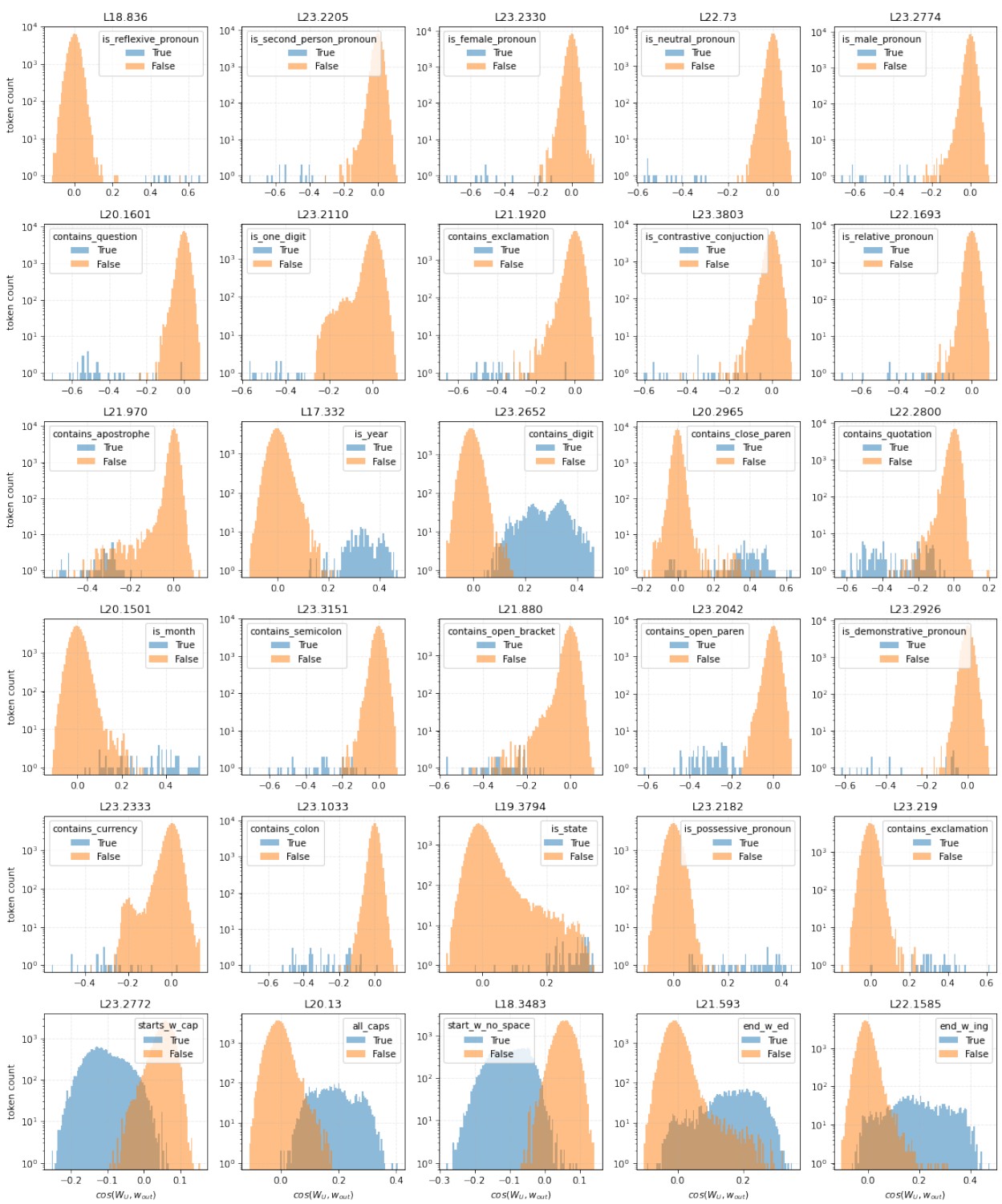

Figure 24: Universal prediction neurons in GPT2-medium-a.

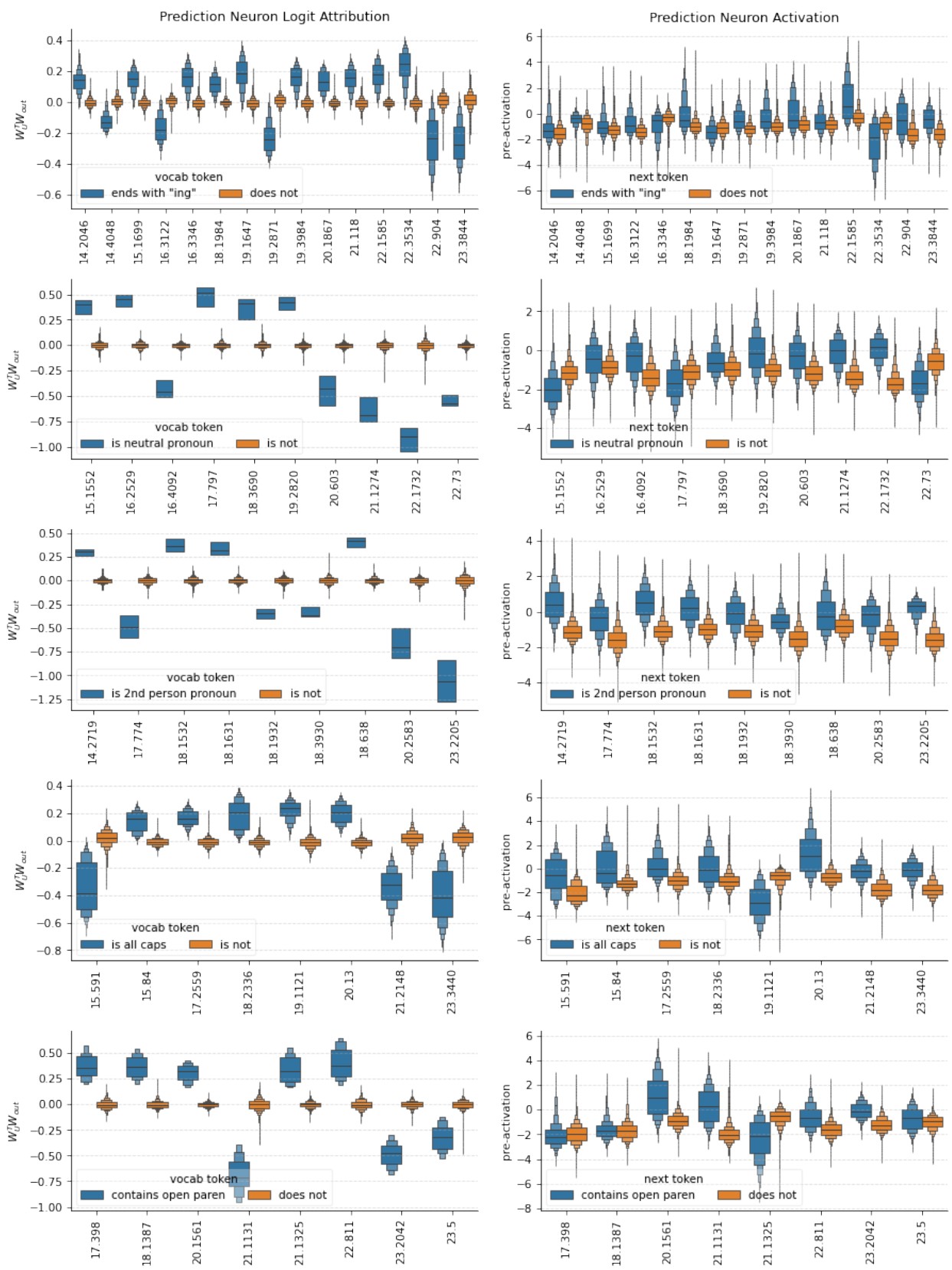

Figure 25: Prediction neurons for the same feature in GPT2-medium-a. Left column depicts logit effect broken down by vocabulary item per neuron and right column shows activation value broken down by true next token per neuron.

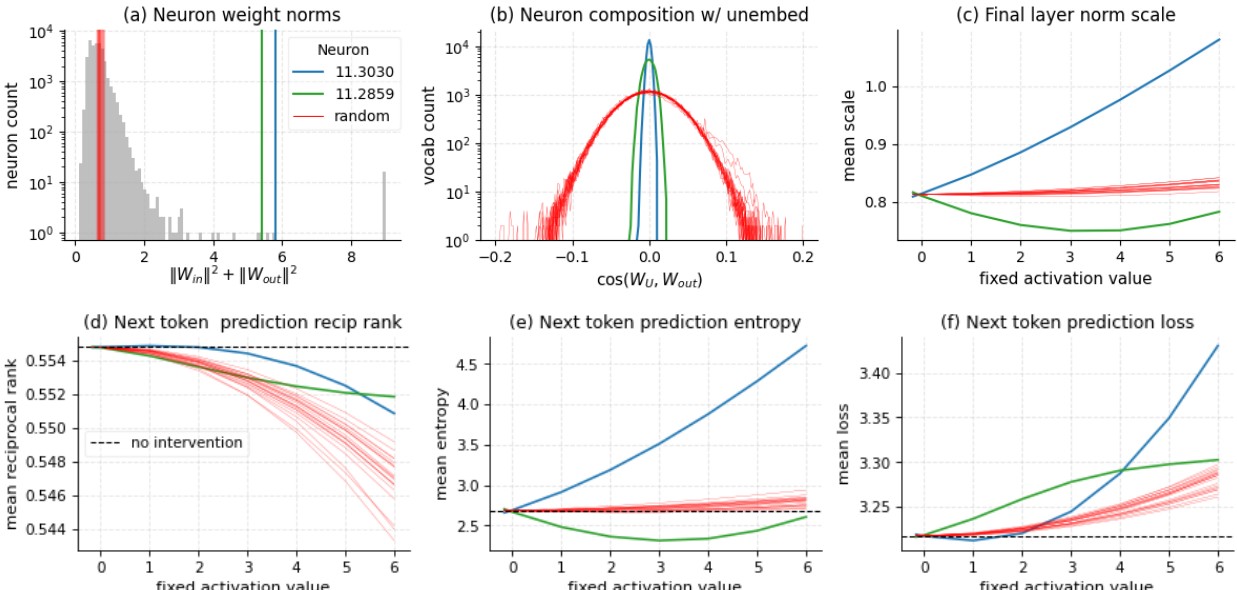

Figure 26: Summary of (anti-)entropy neurons in GPT2-small-a compared to 20 random neurons from final two layers. Entropy neurons have high weight norm (a) with output weights mostly orthogonal to the unembedding matrix (b). When activated, this causes the final layer norm scale to increase dramatically (c) while leaving the relative ordering over the next token prediction mostly unchanged (d). Increased layer norm scale squeezes the logit distribution, causing a large increase in the prediction entropy (e; or decrease for anti-entropy neuron) and an increase or decrease in the loss depending on the model's baseline level of under- or over-confidence (f). Legend applies to all subplots.

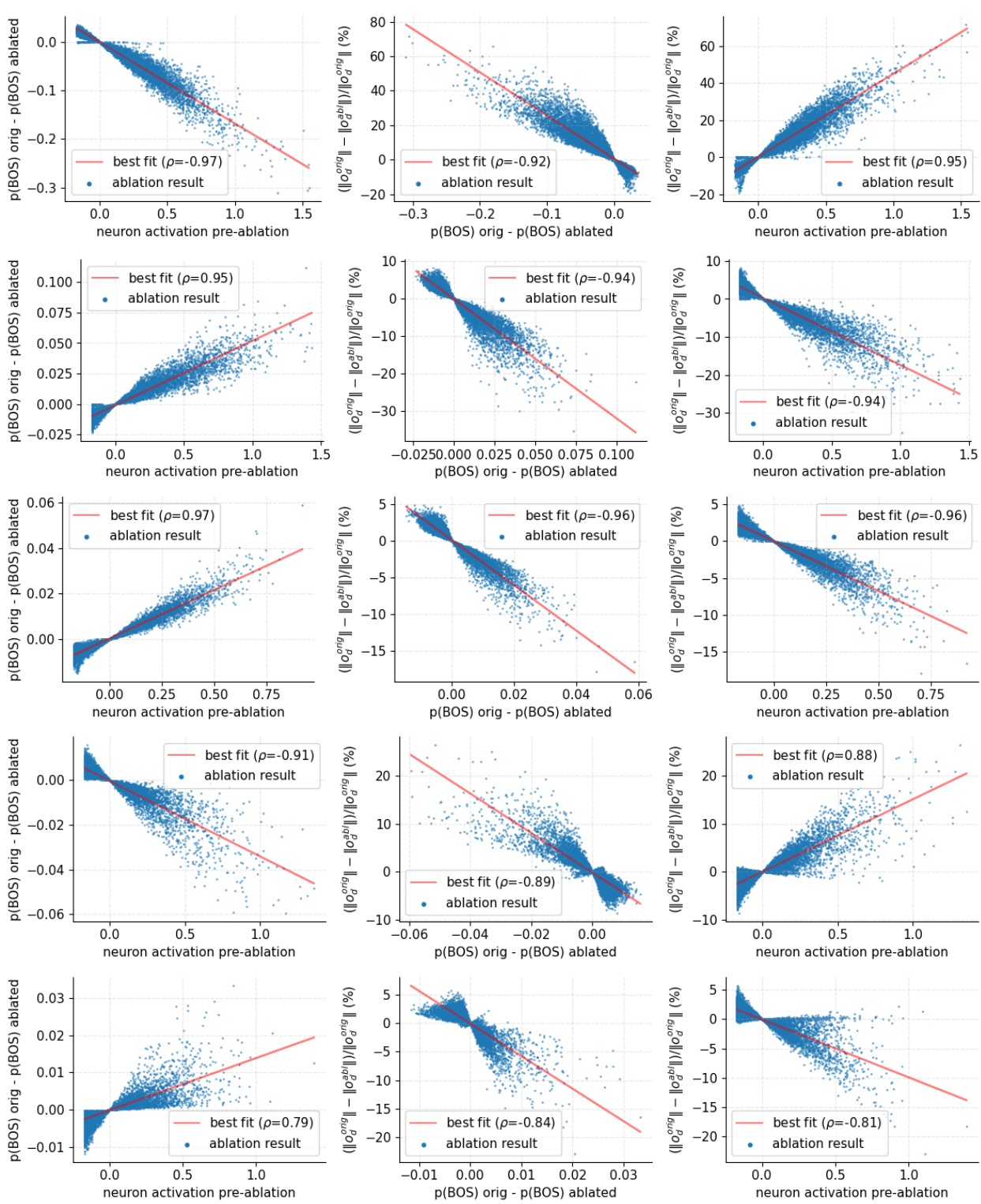

Figure 27: Further examples of attention activation and deactivation neurons. Row 1: A15H8 with L14N411, Row 2: A15H8 with L14N2335, Row 3: A15H8 with L14N1625, Row 4: A20H4 with L19N2509, Row 5: A22H7 with L20N2114

