# OpenReview forum: "Universal Neurons in GPT2 Language Models"
_TMLR — Accepted by TMLR_

### Review · Reviewer_wZeq · 2024-03-05

**Summary Of Contributions:**

This paper explores mechanistic interpretability to understand the universality of individual neurons acrorss GPT-2 model. More specifically, authors compute the pairwise correlations of neuro activations over 100 million tokens. The observations indicate that 1-5% of neurons are universal. By analyzing these patterns in these neuron weights, this paper attributes these neurons to several universal functional roles.

**Audience:**

Yes

**Broader Impact Concerns:**

I think this paper does not need any broader impact concerns.

**Claims And Evidence:**

Yes

**Requested Changes:**

1. I suggest authors to provide more explainations to describle the connections between the observations and the conclusions (i.e., the universality of different neurons).

**Strengths And Weaknesses:**

**Strengths**
1. This paper attempts to analyze neurons of GPT-2 models and reveal the properties of neurons to understand whether they are universal or not. It provides various statistical properties of weights and activations of neurons.



**Weaknesses**
1. Some observations seem weird, why these observations can be attributed to the universality of neurons. The selection of hyperparameters seems specific. Are these hyperparamters (e.g., $\varrho > 0.5$) chosen based on experiences?
2. In these analyses of this paper, authors have summarized many different roles of neurons (e.g., unigram neurons, alphabet neurons, and etc.). But are there any specific rules to identify these neurons and can these obsevations be generalize to other language models?

---

> ### Author Response · Authors · 2024-05-02
> **Response to Reviewer wZeq**
>
> Thank you for taking the time to review our paper. Regarding the weaknesses:
> > Some observations seem weird, why these observations can be attributed to the universality of neurons.
>
> Without knowing more specifically which observations you are referring to, we aren’t sure how to justify them. In general though, we wouldn’t say our results are “attributed” to the existence of universal neurons, but rather “we found lots of universal neurons, and empirically they have particular properties of interest.”
>
> > The selection of hyperparameters seems specific. Are these hyperparamters (e.g., \rho) chosen based on experiences?
>
> The choice of some of these hyperparameters are indeed somewhat arbitrary. We find it unsatisfying to call a neuron with excess correlation 0.51 universal but one with 0.49 not universal, but we had to choose a cutoff. In Figure 11.c, we include a distribution of the excess correlations to give a sense of how many neurons would be deemed universal at different cutoffs. In general, it is the case that universality is a spectrum, as it is really just measuring the degree of similarity.
>
> >In these analyses of this paper, authors have summarized many different roles of neurons (e.g., unigram neurons, alphabet neurons, and etc.). But are there any specific rules to identify these neurons and can these obsevations be generalize to other language models?
>
> As discussed in section 4.3, once we did an initial manual pass of a subset of the neurons, we designed several hundred automatic tests to automatically classify them based on the reduction in variance when conditioning on the test classification. These tests can also be applied to other language models, though these are sometimes complicated by differing tokenizations.
>
> Regarding whether our results generalize, we replicated several of our results on Pythia. In particular the main correlation experiment (Figure 13) as well as the suppression following prediction neuron motif (Figure 22), with the findings being basically the same.
>
> There have also been similar results reported in the literature for other language models, eg. context neurons occurring in pythia models (Gurnee et al. 2023) and position and n-gram neurons in OPT models (Voita et al. 2023).

---

### Review · Reviewer_xxZ1 · 2024-03-09

**Summary Of Contributions:**

This work presents an analysis of the prevalence of 'universal neurons' (neurons with similar response properties) across different instances of GPT-2 trained from different random initializations, together with an analysis of the interpretability of these neurons. The results suggest that a small number of neurons are universal, but that many of these neurons have interpretable functions.

**Audience:**

Yes

**Broader Impact Concerns:**

There are no discernible negative ethical implications of this work.

**Claims And Evidence:**

Yes

**Requested Changes:**

I would appreciate some further insight on the issues raised in the 'weaknesses' section above.

**Strengths And Weaknesses:**

## Strengths
- The paper is clearly written and related work is adequately covered.
- The primary analysis of 'universality' is straightforward and easy to understand, the results are presented in an intuitive manner.
- A particularly compelling finding is that similar neurons emerge in similar layers across models.
- A wide range of functional properties are identified and extensively characterized, including many neuron types that are new discoveries of this work (to my knowledge). This body of findings will be very useful for informing future work on interpretability.

## Weakness
- It is not clear why there should be any relationship between universality and monosemanticity. No justification is provided to motivate this hypothesis.
- It is also unclear why universality should be defined at the level of individual neurons. We can imagine a scenario where a particular layer in two separate networks represents the input in a similar manner, but with representations that are rotated with respect to one another. I don't see why in particular the dimensions of such a representation should align with individual neurons, but it still seems like it would be reasonable to characterize such a scenario as involving a universal solution.
- Many neuron types are considered, with lots of data illustrating examples of these neurons. It is not always clear to what extent these identified neurons are universal though. It would be helpful if, for each neuron type, there was some statement as to whether that neuron type appears in all five models, and in what proportion.
- It would also be helpful to quantify the percentage of the neurons defined as universal end up being interpretable according to the various categories identified in the paper. In the end, there does not seem to be a systematic conclusion regarding the hypothesized relationship between universality and monosemanticity.

---

> ### Author Response · Authors · 2024-05-02
> **Response to Reviewer xxZ1**
>
> We appreciate your careful reading of our paper and the thoughtful comments. Regarding the weaknesses you raised:
> > It is not clear why there should be any relationship between universality and monosemanticity.
>
> The intuition for the connection comes from the Toy Models of Superposition paper (Elhage et al. 2022). They showed that important features (those that are constructed to have maximum effect on the loss) are the features which get their own dedicated neuron (making the neuron monosemantic). Since these GPT2 models are training on the same data, the important features should be similar across the networks. So if monosemantic neurons primarily represent important features, and the important features are shared (ie, universal) across the networks, then we would expect the neurons which are universal across the networks to be representing the same set of important features.
>
> Another line of argument is to consider the probability of a polysemantic neuron being universal. It is extremely unlikely that there would exist a pair of neurons which activates for the same k >> 0 unrelated features, assuming unrelated features have roughly similar probabilities of being assigned to neurons. Moreover, this simple model suggests that as k -> 1 it is much more likely for there to exist a neuron which represents the same set of features.
>
> > It is also unclear why universality should be defined at the level of individual neurons. We can imagine a scenario where a particular layer in two separate networks represents the input in a similar manner, but with representations that are rotated with respect to one another. I don't see why in particular the dimensions of such a representation should align with individual neurons, but it still seems like it would be reasonable to characterize such a scenario as involving a universal solution.
>
> We agree that the case you describe should also be considered universal. As we discuss in section 3.1, there are many potential notions of universality, and the neuron universality we study here is a particularly strong version of implementation universality.
>
> We study this notion for two reasons. (1) To help determine to what extent the neuron basis is a good unit of analysis. We would expect that a representation space which exhibits higher levels of universality to be better all things equal. (2) Because it constitutes something of a bound on weaker notions of universality. That is, to the extent that changing random seed is a “smaller” change than swapping architectures or data distributions, we would expect the measured universality to be even less in these instances.
>
> Based on our results, we think the most natural future work is to replicate these experiments using dictionary learning (Bricken et al. 2023) to pull out the underlying features, and to test if these features are the same across models. Bricken et al. found much higher levels of correlation between features than neurons in a pair of 1 layer models, but it is important to verify this is true in a less toy setting.
>
> > It is not always clear to what extent these identified neurons are universal though. It would be helpful if, for each neuron type, there was some statement as to whether that neuron type appears in all five models, and in what proportion.
>
> Every single neuron presented in the main text had an excess correlation greater than 0.5 (with the exception of the entropy neurons which had excess correlation 0.45) so they all appeared in all models (as briefly mentioned in 4.3).
>
> >  It would also be helpful to quantify the percentage of the neurons defined as universal end up being interpretable according to the various categories identified in the paper.
>
> It was initially our intention to include a table reporting these numbers. However, we realized that it was difficult to get good numbers for two reasons
> 1) Explained variance ratio gives you a scalar rather than a classification, and you again need to define a threshold. This means any reported numbers are very sensitive to the overall threshold.
> 2) For some categories our tests are quite inadequate (like semantic neurons), and would substantially undercount the total number.
>
> In other words, any numbers we report would be a reflection of the threshold and our test coverage more than anything about the model, so we instead chose to report the top 20/30 neurons from each category in the appendix, to at least prove the existence.

---

> > ### Comment · Reviewer_xxZ1 · 2024-05-29
> > **Reply**
> >
> > Thanks very much to the authors for these responses. I now understand better the motivation for proposing a link between universality and monosemanticity, but it still seems quite likely to me that there will be universal aspects of learned representations that will not be reflected in the response profiles of individual neurons, but instead will be better understood at the population level.
> >
> > Regarding the potential for a more systematic comparison of universality and monosemanticity, in order to address the issue of threshold selection, one approach might be to perform an ROC/AUC analysis, or to use a bias-free measure such as d'.

---

### Review · Reviewer_v8MV · 2024-04-19

**Summary Of Contributions:**

The paper addresses the question whether there are universal neurons in Transformer-based language models. In the authors' definition of universality, a neuron is universal if in a second model of the same architecture but trained with different initial conditions there exists another neuron that behaves largely in the same way, i.e. has highly correlated activation patterns over a large number of input sequences. The authors find a relatively small subset of such "universal" neurons and investigate their properties.

**Audience:**

Yes

**Claims And Evidence:**

Yes

**Requested Changes:**

I would like the authors to improve the clarity on the points listed above, include a discussion on how we use the insights from this analysis and fix the invalid conclusion.

**Strengths And Weaknesses:**

### Strengths

 1. Addresses an important question: at what level can we "understand" a neural network?
 1. Reports a number of interesting observations about the inner workings of Transformers
 1. Tests two different models (GPT2 and Pythia), observing similar results



### Weaknesses

 1. Reasoning is not always easy to follow
 1. Observations are largely descriptive; not clear how to derive insights for future improvements


### Detailed comments


Overall the paper is well organized and mostly well written. It reports a number of (not necessarily novel) observations about the inner workings of Transformers that researchers interested in interpretability methods would most likely find interesting and could inspire future work. I am therefore overall fairly supportive of the paper, although I believe it could be improved by more clarity in the exposition of the results and by more clearly stating if there are any implications for future neural network design that can be derived from this work.


### 1. Clarity

While I find the overall organization of the paper easy to follow, I got lost a few times in the individual sections. In particular, the following sections' clarity could be improved:

 1. Section 4.1: While I can generally follow the reasoning and analysis, I am a bit puzzled about the results of Fig. 2b: Doesn't the fact that the difference between max-max and min-max is larger for "universal" neurons imply that such neurons are more likely to be missing in at least one of the models (the min-max is smaller than expected by the fluctuations across most neurons), i.e. they're not that universal?

 1. Section 4.2: It is not obvious to me what cos(w_out, w_U) measures. As far as I understand W_U is used only in the last layer. Why would it be interesting to look at this direction in earlier layers? The authors could provide a better rationale for this analysis.

 1. Section 4.3: I did not understand the significance of Eq. (4). The variables y and beta are unclear to me and I'm not sure what the quantity actually measures and how it exactly relates to the authors "finding" neurons of the types described below. Perhaps this part can be unpacked a bit more?

 1. Section 5.1: Again W_U shows up in an analysis applied to all layers, but -- unless I'm confused -- it never gets applied to earlier layers. Thus, I am not sure what's the meaning of or the rationale behind this analysis. Also, I did not directly understand why the product of w_out and W_U? Isn't there also the residual path that gets added and layer norm in between? What does that product tell us? I think this could be explained better and perhaps supported by an illustration (illustrations may help also for earlier points).

 1. Section 5.1: I could not follow the paragraph starting with "When studying the activations of suppression neurons..." at all. It states that they "activate far more often when the next token is in fact from the set of tokes they express". Far more often than what? And if they indeed activate more often, why are they called suppression neurons?

 1. Section 5.3: It wasn't immediately clear to me why "the vector W_O v_BOS is constant for all prompts". Is it because the causal attention mask prevents this token from attending to any other token?



### 2. What do we learn?

Although I find the observations overall interesting, I am not really sure what we learn from this type of "neuroscience" experiments on deep learning models. They describe what some neurons do, but what follows from such knowledge? Can we use it to design better models? If so, how? Can we use it to understand why they fail if they fail? If so, how? I think it would be great if the authors could discuss a bit more how they picture the path forward.



### 3. Invalid conclusion

I don't think the following conclusion in the discussion is valid: "... find that only about 1-5% of neurons are universal across models, constituting another piece of evidence that individual neurons are not the appropriate unit of analysis for most network behaviours." -- The number of universal neurons is completely arbitrary. As you state yourself, there is no objective threshold and you arbitrarily chose 0.5. Had you chosen 0.2, most of the neurons would be "universal" and the conclusion would have been the opposite. In fact, I do not see evidence in this paper for the claim that "individual neurons are not the appropriate unit of analysis for most network behaviours", so this claim should be removed or rephrased.



### Typos:

 - P. 4, line 5 from bottom: "in terms of [the] its"
 - P. 11, last sentence in 5.2 "... models [something missing] the second ..."
 - P. 12, line 5: "Positive scores suggest[s] activation..."

---

> ### Author Response · Authors · 2024-05-02
> **Response to Reviewer v8MV (1/2)**
>
> We appreciate your careful reading of our work and for the many recommendations to improve the clarity of our paper!
>
> We have updated the manuscript to improve readability, but to respond to the points (and provide additional explanation):
> > Fig. 2b: Doesn't the fact that the difference between max-max and min-max is larger for "universal" neurons imply that such neurons are more likely to be missing in at least one of the models (the min-max is smaller than expected by the fluctuations across most neurons), i.e. they're not that universal?
>
> The issue here is that the correlation range is much smaller for non-universal neurons, and so it is a somewhat misleading comparison. Specifically, from Fig 2.a we see that the mode is around 0.25 for mean-max correlation (note the log scale), so the vast majority of neurons will have correlation around this range, which is largely just from randomness. But since the nominal correlation range for universal neurons is ~0.65-1, we should a priori expect more variance.
>
> >Section 4.2: It is not obvious to me what cos(w_out, w_U) measures. As far as I understand W_U is used only in the last layer. Why would it be interesting to look at this direction in earlier layers? The authors could provide a better rationale for this analysis.
>
> Thank you for raising this. We provide more justification in 5.1, but the intuition is that this measures how much of a direct effect these neurons have on the next token probability, by directly changing the logits. W_U is applied at the end of the network, but it is applied to the residual stream, which is the sum of all previous layers.  This is a fairly standard technique; see the following references for more details:
> Dar, Guy, et al. "Analyzing transformers in embedding space." arXiv preprint arXiv:2209.02535
> Geva, Mor, et al. "Transformer feed-forward layers build predictions by promoting concepts in the vocabulary space." arXiv preprint arXiv:2203.14680 (2022).
>
> We added a note in 4.2 to clarify this to unfamiliar readers
>
> > Section 4.3: I did not understand the significance of Eq. (4). The variables y and beta are unclear to me and I'm not sure what the quantity actually measures and how it exactly relates to the authors "finding" neurons of the types described below. Perhaps this part can be unpacked a bit more?
>
> Indeed this was a bit terse. We are essentially comparing the variance of the original activation distribution to the average variance of the two distributions (weighted by how many samples there are, which is the purpose of beta) when split by the explanation (ie, whether y_i=1 or y_i=0). To give a concrete case, if we look at 4.b, there is a lot of variance in the overall distribution, but if we condition on the y=inifinite_verb explanation, each of the blue and orange distributions have much less variance than the bimodal distribution. Of course, most tokens are not infinitive verbs, so we should weight the new variance in proportion to the number of samples in each distribution (beta=fraction of infinitive verbs).
>
> As another source of intuition and justification, this is exactly the formula for selecting a split in a regression tree (where y_i gives the indicator of which samples are on what side of the split).
>
> We added an additional sentence to unpack this further.
>
> > Section 5.1: Again W_U shows up in an analysis applied to all layers, but -- unless I'm confused -- it never gets applied to earlier layers. Thus, I am not sure what's the meaning of or the rationale behind this analysis. Also, I did not directly understand why the product of w_out and W_U? Isn't there also the residual path that gets added and layer norm in between? What does that product tell us? I think this could be explained better and perhaps supported by an illustration (illustrations may help also for earlier points).
>
> The first paragraph of section 5.1 introduces the idea of direct logit attribution, but to expand a bit:
> W_U is never applied to a “layer,” it is applied to the final residual stream state, which is the sum of all previous layers. The output of an MLP layer is the linear combination of all neuron output weights w_out weighted by the neuron activations. Therefore, there exists a direct path from each individual neuron to changing the logit distribution, given by W_U w_out, since the final residual state is \sum_layer \sum_neurons w_out * activation (plus the embeddings and attention). This does ignore the nonlinear effect of layernorm, but the assumption is that an individual neuron is unlikely to change the layer norm scale very much, and therefore we can treat it as being a constant scale factor (see the appendix “Handling Layer Normalization” in Elhage et al. 2021 “A Mathematical Framework for Transformer Circuits” for additional details) .
>
> In short, the product tells us what the direct effect on the next token probabilities are when the neuron has activation one (regardless of layer). We added a bit more clarity in the text.

---

> ### Author Response · Authors · 2024-05-02
> **Response to Reviewer v8MV (2/2)**
>
> >Section 5.1: I could not follow the paragraph starting with "When studying the activations of suppression neurons..." at all. It states that they "activate far more often when the next token is in fact from the set of tokes they express". Far more often than what? And if they indeed activate more often, why are they called suppression neurons?
>
> Thank you for pointing out the lack of readability here. Put simply, a suppression neuron is a neuron which decreases the probability of a token class (eg, all tokens containing an open parenthesis as in Fig 5.2), but we observe that suppression neurons activate much more frequently when the next token actually contains an open parenthesis than when it doesn’t. They are called suppression neurons because they have the effect of lowering the next token probability when active.
>
> We have reworded this paragraph to improve its readability.
>
> >Section 5.3: It wasn't immediately clear to me why "the vector W_O v_BOS is constant for all prompts". Is it because the causal attention mask prevents this token from attending to any other token?
>
> Yes, since the BOS token is always the first in the sequence, there is no context, so all of the activations are always the same. We added a note clarifying.
>
> > They describe what some neurons do, but what follows from such knowledge? Can we use it to design better models? If so, how? Can we use it to understand why they fail if they fail? If so, how? I think it would be great if the authors could discuss a bit more how they picture the path forward.
>
> Our motivation here is primarily to improve our understanding of models for interpretability sake, rather than make models more performant. Specifically, we think interpretability is largely an immature field, without much grounding theory. Therefore, we think it's inherently valuable to gain a lot of “empirical surface area” on real networks to constrain the hypothesis space and develop the theory and practice of interpretability.
>
> Example insights from our paper that might help future interpretability researchers
> - Single neuron ablations can be misleading if there is an ensemble of similar neurons or there are neurons which consistently cancel each other out.
> - The depth specialization results and the prediction followed by suppression neuron transition emphasize the sequential and residual nature or model processing.
> - Our results on entropy neurons exemplify a concrete case where it wouldn’t be valid to linearize layer norm.
> - Our results on attention deactivation neurons show how certain features might be difficult to interpret, because they are features which take internal actions rather than represent external inputs.
>
> > I don't think the following conclusion in the discussion is valid: "... find that only about 1-5% of neurons are universal across models, constituting another piece of evidence that individual neurons are not the appropriate unit of analysis for most network behaviours." -- The number of universal neurons is completely arbitrary. As you state yourself, there is no objective threshold and you arbitrarily chose 0.5. Had you chosen 0.2, most of the neurons would be "universal" and the conclusion would have been the opposite. In fact, I do not see evidence in this paper for the claim that "individual neurons are not the appropriate unit of analysis for most network behaviours", so this claim should be removed or rephrased.
>
> We agree that the threshold makes claiming a specific number of universal neurons somewhat unprincipled. However, we still think it is unreasonable to view our results as providing evidence for widespread universality of the neuron basis. To conclude that neurons are not the right unit of analysis, we agree that we would likely need to show that it is less universal compared to sparse autoencoders (similar to Bricken et al. 2023).  Hence we have removed the unit of analysis claim and have qualified the number of universal neurons claim by noting the threshold.
>
> > Typos
>
> Thank you for catching these! They have been fixed.

---

### Author Response · Authors · 2024-05-02
**Minor revision**

We thank all of the reviewers for their helpful comments. We have uploaded an edited manuscript (with changes in blue) in response to the reviewer feedback.

---

### Decision · Action_Editor_GTdM · 2024-06-07

**Recommendation:** Accept with minor revision

**Comment:**

The reviewers felt that the clarity of the paper could still be greatly improved. The AE strongly encourages the authors to try to address all of the points regarding clarity provided by the reviewers. As well, there was still a request from one reviewer to include a quantitative analysis comparing universality and monosemanticity (using an approach that doesn't require setting an arbitrary threshold, such as ROC/AUC analysis). The AE would recommend that the authors also provide this.

**Audience:**

Yes, the TMLR audience would be interested in this paper, as it shows an interesting empirical phenomenon that speaks to potential shared properties across model instantiations.

**Claims And Evidence:**

In this paper the authors examine whether there are neurons with the same selectivity properties that consistently emerge in different trained networks, and whether these "universal neurons" are more interpretable in their activity patterns than other neurons. To examine this, the authors train multiple GPT-2 models and examine correlations between the neurons' activations across many tokens. They report the existence of a small percentage of neurons (1-5%) that show consistent activation patterns (high correlations) across trained models, i.e. "universal neurons". The authors examine the properties of these neurons, and argue that they have more interpretable activations and weights.

The reviewers had a number of questions, particularly around clarity, motivations, and method details. Based on the authors' responses, the reviewers were largely satisfied that the claims of the authors had been substantiated. However, there were some remaining recommendations that should be included (see below).

---

> ### Author Response · Authors · 2024-06-16
> **Camera Ready Revision**
>
> Thank you for the decision.
>
> In addition to the previous revision which made a number of clarity improvements in response to reviewers v8MV and xxZ1 (marked in blue in the previous version), we have made included more revisions for the camera ready.
>
> We added a new discussion section in the appendix (A) which addresses in detail the proposed hypothesis of the connection between universality and monosemanticity. Additionally we included the discussion of how we see our work as being useful (emphasizing the usefulness to interpretability rather than capabilities).
>
> To address the thresholding, we added Figure 14 which shows the (complement) cumulative distribution of our various correlation metrics. This way, it is possible to read off the fraction of neurons that are deemed universal under any threshold.